# Test-Time Selective Adaptation for Uni-Modal Distribution Shift in Multi-Modal Data

MingCai Chen [* 1]  Baoming Zhang [* 2]  Zongbo Han [3]  Wenyu Jiang [2]  Yanmeng Wang [1]  Shuai Feng [2]
Yuntao Du [4]  Bingkun Bao [1]

## Abstract

Modern machine learning applications are characterized by the increasing size of deep models and the growing diversity of data modalities. This trend underscores the importance of efficiently adapting pre-trained multi-modal models to the test distribution in real time, i.e., multi-modal test-time adaptation. In practice, the magnitudes of multi-modal shifts vary because multiple data sources interact with the impact factor in diverse manners. In this research, we investigate the the under-explored practical scenario *uni-modal distribution shift*, where the distribution shift influences only one modality, leaving the others unchanged. Through theoretical and empirical analyses, we demonstrate that the presence of such shift impedes multi-modal fusion and leads to the negative transfer phenomenon in existing test-time adaptation techniques. To flexibly combat this unique shift, we propose a selective adaptation schema that incorporates multiple modality-specific adapters to accommodate potential shifts and a "router" module that determines which modality requires adaptation. Finally, we validate the effectiveness of our proposed method through extensive experimental evaluations. Code available at https://github.com/chenmc1996/Uni-Modal-Distribution-Shift.

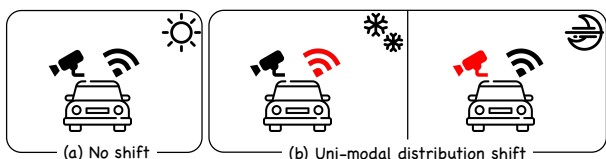

*Figure 1.* A self-driving car equipped with complementary camera and LiDAR sensors navigating under different conditions. In the right figure, the icy road surface primarily impacts the LiDAR signals, while poor light conditions mainly affect the camera signals.

## 1. Introduction

In recent years, multi-modal learning has emerged as a crucial area in both academic research and practical applications due to its remarkable ability to process a wide range of data types (Huang et al., 2021; Xu et al., 2023). In the open-world environment across diverse applications, distribution shifts constantly occur and significantly degrade the performance of static models (Liang et al., 2024; Shi et al., 2024; Han et al., 2023). Existing research on distribution shifts, especially in multi-modal data, often implicitly assumes a "global" distribution shift where all modalities experience distribution changes. However, many real-world corrupting factors impact only specific modalities, e.g., light changes can solely shift the distribution of camera signals, while the LiDAR data remains immune as shown in Fig. 1. For the first time, we term this type of shift *uni-modal distribution shift*, which is distinct from the previously considered global distribution shifts. Nevertheless, this setting is of great practical significance. In many cases, complementary data modalities are often preferred, yet they are likely to be subject to different disturbances. We theoretically show how the uni-modal distribution shift undermines the attention-based multi-modal model in Sec. 3.3.

Methodologically, to address distribution shifts, Test-Time Adaptation (TTA), which involves continuously updating models to adapt to the shifted data distribution in test environments, has become a cutting-edge approach (Wang et al., 2020; Cao et al., 2025; Yang et al., 2024). Therefore, our work also focuses on utilize TTA to solve the uni-modal distribution shift problem. Subsequently, we investigate the adequacy of existing TTA methods for addressing the

---

*Equal contribution [1] Nanjing University of Posts and Telecommunications [2] State Key Laboratory for Novel Software Technology at Nanjing University, Nanjing University [3] College of Intelligence and Computing, Tianjin University [4] Joint SDU-NTU Centre for Artificial Intelligence Research (C-FAIR) & School of software, Shandong university. Correspondence to: Yuntao Du <yuntaodu@sdu.edu.cn>, Bingkun Bao <bingkunbao@njupt.edu.cn>.

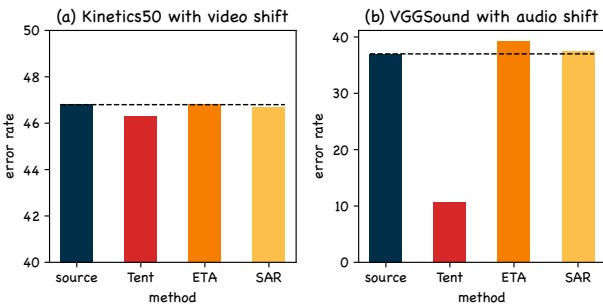

*Figure 2.* TTA methods' performance on uni-modal distribution shift. The introduction of adaptation techniques on the unshifted test data results in limited performance gain or even degeration (Pre-trained model (Source) vs. Tent, ETA, and SAR).

uni-modal distribution shifts. Recent TTA and multi-modal TTA techniques—such as Tent (Wang et al., 2020), ETA (Niu et al., 2022), SAR (Niu et al., 2023), and READ (Yang et al., 2024)—indifferently perform adaptation on every modality-specific encoder or the modality fusion module utilizing techniques like batch re-normalization or entropy minimization (detailed in Sec. 2). However, when distribution shift does not occur in all modalities, these approaches lead to redundancy or even negative effects, resulting in overfitting in unshifted modalities to which the original model already generalizes. We perform experiments on multi-modal datasets with uni-modal distribution shift, and discovery limited performance gain. Tent and SAR even exhibit the negative transfer phenomenon (Rosenstein et al., 2005), as shown in Fig. 2. In short, uni-modal distribution shift presents a unique challenge that common TTA methods struggle to address: *How to flexibly adapt to the shift in any modality without harming the multi-modal fusion of other unshifted modalities?*

To achieve this goal, we propose a straightforward yet effective approach for flexible modality-specific adaptation. It mainly uses a learnable "router" that determines which modality should be adapted. On top of that, we introduce a lightweight adapter for each modality's feature representations to accommodate potential distribution shifts at test time. By allowing the router to automatically activate one adapter while disabling others, we achieve flexible selective adaptation. All components are updated in an end-to-end manner during test time using simple self-training techniques. Consequently, our method effectively enables safe TTA for uni-modal distribution shifts in multi-modal data. It is verified on multi-modal datasets, namely Kinetics50 (Kay et al., 2017) and VGGSound (Chen et al., 2020), with 21 types of uni-modal distribution shifts across different modalities. The main contributions of this work are:

- We identify the unique challenges of uni-modal shift in multi-modal data through theoretical analysis (i.e., large fluctuations in cross-modal attention) and empiri-

cal analysis (i.e., negative transfer).

- We propose a simple but effective method using lightweight adapters for each modality's feature representation. A learnable "router" is designed to automatically activate the adapter for the shifted modality and disable the adapters for unshifted modalities.

- Our method is validated through extensive experiments on the uni-modal distribution shifted datasets, and the results show that our approach achieves superior performance. The comprehensive experimental setup guarantees the robustness and generalizability of our method.

## 2. Related Works

### 2.1. Test-Time Adaptation

Distribution shift easily happens between training and test data. Meanwhile, adaptation to the test distribution is limited by various factors, with the most prominent ones being the lack of test data labels and the limited computational resources for further tuning. TTA enables flexible, online adaptation to the current test distribution, attracting increasing attention in the field.

Early efforts (Wang et al., 2020; Lim et al., 2023) posit that the statistics in the batch normalization layers embody distribution knowledge. For a mini-batch of data $x$ in the middle batch normalization layer, the activation $\hat{x}$ is given by: $\hat{x} = \frac{x - \mathbb{E}[X]}{\sqrt{\mathbb{V}[X]}} \cdot \gamma + \beta$, where $\mathbb{E}[X]$ and $\mathbb{V}[X]$ are the estimated mean and variance over the data distribution (For illustration, we omit the constant for numerical stability). During test-time, the mean and variance are set as the moving average of training batches' statistics. Here, $\gamma$ and $\beta$ are learnable scale and shift parameters. The re-normalization technique is used to adapt to the new distribution. During test-time, instead of fixing the running statistics (mean $\mu_{run}$ and standard deviation $\sigma_{run}$) computed during training, Adabn (Li et al., 2017) recomputes the batch statistics for every test batch. The adjusted batch normalization formula for test data $x$ becomes: $\hat{x} = \frac{x - \mu_{run}}{\sigma_{run}} \cdot \gamma + \beta$. This way, it utilizes the statistics from the test distribution. In contrast, TENT (Wang et al., 2020) optimizes the affine parameters $\gamma$ and $\beta$ in batch normalization layers by minimizing the entropy of the prediction probability. ETA (Niu et al., 2022) excludes unreliable and redundant samples from the entropy minimization calculation. SAR (Niu et al., 2023) performs sample selection from the gradient aspect, removing samples that cause large gradients and encouraging model weights to reach a flat minimum. However, most of the TTA methods do not consider multi-modal scenarios. These methods can typically be deployed, for instance, in the encoder of each modality or the multi-modal fusion module. But in the context of the new uni-modal distribution shift within

multi-modal data, these methods would suffer from negative transfer issues.

## 2.2. Test-Time Adaptation for Multi-Modal Data

Multi-modal learning has emerged as a promising approach for understanding the world through different data modalities. Moreover, publicly available large-scale pre-trained multi-modal models have been widely applied. However, in the open world, the dynamic and continuously changing data distribution damages the applicability (Sehwag et al., 2019; Zhou, 2022; Cao et al., 2023; Xiong et al., 2024; Cao et al., 2025). Thus, there is an urgent need for the adaptation of corresponding TTA techniques to efficiently adapt to new multi-modal data. In MM-TTA (Shin et al., 2022), a co-training-style (Blum & Mitchell, 1998) intra-modal pseudo-labeling, followed by inter-modal pseudo-label refinement, is introduced for 3D semantic segmentation. To bridge the gap in distribution shift between modalities. It maintains two norm statistics: one is directly updated by the test data, another is slowly updated with a momentum from the direct-updated batch norm parameter. The work focuses on the situation where distribution shifts occur within all modalities, yet adaptations should be separately adjusted. Compared with our method, it manually assigns different degrees of shift to specific modalities. In contrast, our method performs dynamic routing in an end-to-end and learnable manner. READ (Yang et al., 2024) investigates multi-modal shifts and includes preliminary experiments on uni-modal shifts. It identifies the "reliability bias" problem, indicating that when certain modalities are affected by distribution shifts, the information discrepancies between modalities are amplified. To address this, READ modulates the attention-based fusion layers. However, rather than reducing attention on the shifted modality, we aim to fully extract information from them through modality-specific adaptation, which utilizes the shifted data more efficiently.

## 3. Method

### 3.1. Problem Definition of Uni-Modal Distribution Shift

In multi-modal learning, data is represented through diverse modalities such as text, images, and audio. Uni-modal distribution shift occurs when the distribution of only one modality changes between the source and target domains, while the others remain unchanged. Notably, we do not know which modality the shift occurs in. This presents unique challenges as the model must perform selective adaptation. Its formalization is: The joint distribution of the multi-modal source domain is given by $P_S(\boldsymbol{x}^{(1)}, \boldsymbol{x}^{(2)}, \ldots, \boldsymbol{x}^{(M)}, Y)$. For the target domain, it is $P_T(\boldsymbol{x}^{(1)}, \boldsymbol{x}^{(2)}, \ldots, \boldsymbol{x}^{(M)}, Y)$. Here, the marginal distribution of the $k$-th modality in the target domain is shifted, i.e., $P_T(\boldsymbol{x}^{(k)}) \neq P_S(\boldsymbol{x}^{(k)})$, while for the other modalities, $P_T(\boldsymbol{x}^{(i)}) = P_S(\boldsymbol{x}^{(i)}), \forall i \neq$

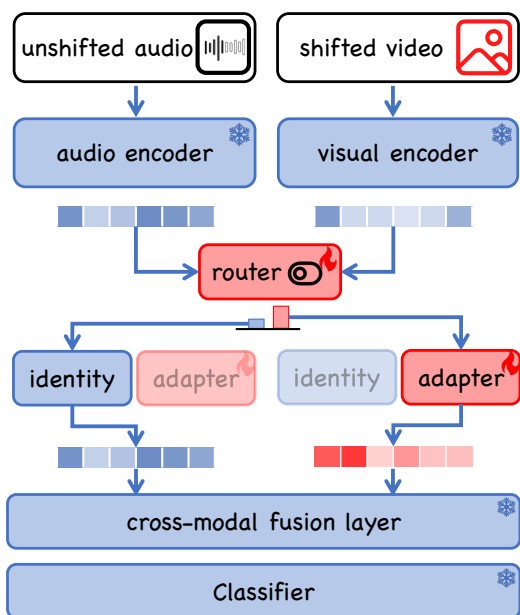

*Figure 3.* The architecture of our model.

$k$. Additionally, the conditional distribution of the target label $Y$ given the input $\boldsymbol{x}^{(1)}, \boldsymbol{x}^{(2)}, \ldots, \boldsymbol{x}^{(M)}$ remains the same, i.e., $P_S(Y|\boldsymbol{x}^{(1)}, \boldsymbol{x}^{(2)}, \ldots, \boldsymbol{x}^{(M)}) = P_T(Y|\boldsymbol{x}^{(1)}, \boldsymbol{x}^{(2)}, \ldots, \boldsymbol{x}^{(M)})$.

In TTA for uni-modal distribution shift, the model is tasked with generating predictions for streaming multi-modal data. Simultaneously, model updates are carried out to adapt to the uni-modal shifted distribution.

### 3.2. Architecture of Pre-trained Model

For simplicity, we will illustrate our approach using two modalities: audio and video. This framework can be easily extended to more general multi-modal cases. Suppose there is a model pre-trained on a labeled multi-modal source dataset. As shown in Fig. 3, the model comprises modality-specific encoders (visual encoder $f^{(v)}$ and audio encoder $f^{(a)}$), a cross-modal fusion layer $f^{(m)}$, and a classifier $f^{(c)}$. Both $f^{(a)}$ and $f^{(v)}$ are transformer encoders (Vaswani et al., 2017) that map the input modality into a series of tokens. The cross-modal fusion module $f^{(m)}$ is mainly a self-attention layer (Vaswani et al., 2017).

### 3.3. Theoretical Analysis

With the problem setting and model architecture established, we initiate on a theoretical analysis aimed at understanding the implications of uni-modal distribution shift on multi-modal fusion. We begin our analysis from the core module of multi-modal fusion: the self-attention mechanism. It plays a pivotal role in multi-modal fusion by aggregating

intra-modal and inter-modal token representations with different weights. Mathematically, it is defined by the formula:

$$\begin{aligned} &\text{Attention}(Q(\boldsymbol{z}), K(\boldsymbol{z}), V(\boldsymbol{z})) \\ &= \text{softmax}\left(\frac{Q(\boldsymbol{z})K(\boldsymbol{z})^T}{\sqrt{d_k}}\right)V(\boldsymbol{z}), \end{aligned} \quad (1)$$

where query $Q(\cdot)$, key $K(\cdot)$, and value $V(\cdot)$ are the linear transformation of the tokens' representation $\boldsymbol{z}$, $d_k$ is the scale factor. The attention logit, a crucial determinant in the self-attention, is defined as the inner product between the query and key matrices prior to the softmax function. Its value depends on the representations of two tokens involved in the computation. Formally, we define the attention logit:

**Definition 3.1** (Attention Logit)**.** The *attention logit* (AL) is defined as the inner product between query and key matrices:

$$\text{AL}(\boldsymbol{z}_i, \boldsymbol{z}_j) = Q(\boldsymbol{z}_i)K(\boldsymbol{z}_j)^T = \boldsymbol{z}_i\left(W^Q W^{K^T}\right)\boldsymbol{z}_j^T, \quad (2)$$

where $\boldsymbol{z}_i \in \mathbb{R}^F$ and $\boldsymbol{z}_j \in \mathbb{R}^F$ are a pair of embedding of tokens involved in the self-attention calculation, and $W^Q, W^K \in \mathbb{R}^{F \times F}$ are the linear projection matrices.

However, the occurrence of a distribution shift in any modality can disrupt this mechanism. Without loss of generality, we analyze the shift in the audio modality: Token representations of the shifted audio are corrupted, and we denote it as $\acute{\boldsymbol{z}}_i^{(a)}$ and $\acute{\boldsymbol{z}}_j^{(a)}$, different from the tokens from unshifted modality $\boldsymbol{z}_k^{(v)}$. The distribution shift would result in the changes of attention logit between two shifted audio tokens: $\text{AL}(\acute{\boldsymbol{z}}_i^{(a)}, \acute{\boldsymbol{z}}_j^{(a)}) - \text{AL}(\boldsymbol{z}_i^{(a)}, \boldsymbol{z}_j^{(a)})$. The change between one shifted audio token and unshifted one video is token: $\text{AL}(\acute{\boldsymbol{z}}_i^{(a)}, \boldsymbol{z}_k^{(v)}) - \text{AL}(\boldsymbol{z}_i^{(a)}, \boldsymbol{z}_k^{(v)})$.

**Proposition 3.2.** *Under the zero-mean additive shift assumption (Kim et al., 2020):*

$$\sup\left(\mathbb{D}\big[\underbrace{\text{AL}(\acute{\boldsymbol{z}}_i^{(a)}, \boldsymbol{z}_k^{(v)}) - \text{AL}(\boldsymbol{z}_i^{(a)}, \boldsymbol{z}_k^{(v)})}_{\textit{change of cross-modal AL after shift}}\big]\right)$$

$$< \sup\left(\mathbb{D}\big[\underbrace{\text{AL}(\acute{\boldsymbol{z}}_i^{(a)}, \acute{\boldsymbol{z}}_j^{(a)}) - \text{AL}(\boldsymbol{z}_i^{(a)}, \boldsymbol{z}_j^{(a)})}_{\textit{change of intra-shited-modal AL after shift}}\big]\right)$$

*when* $\mathbb{E}[\|\boldsymbol{z}_k^{(v)}\|_2^2] < \mathbb{E}[\|\boldsymbol{z}_i^{(a)}\|_2^2] + \mathbb{E}[\|\boldsymbol{z}_j^{(a)}\|_2^2] + \mathbb{E}[\|\varepsilon_j\|_2^2]$

$$(3)$$

*where* $\sup(\cdot)$ *is the least upper bound.* $\varepsilon_j$ *is the noise on token* $\boldsymbol{z}_j^{(a)}$*.* $\mathbb{E}[\|\boldsymbol{z}_k^{(v)}\|_2^2]$, $\mathbb{E}[\|\boldsymbol{z}_i^{(a)}\|_2^2]$, $\mathbb{E}[\|\boldsymbol{z}_j^{(a)}\|_2^2]$, $\mathbb{E}[\|\varepsilon_j\|_2^2]$ *are the expected value of the squared norm of the token from different modalities and noise, respectively.*

*Proof. See the appendix for the formal proof.*

*Remark* 3.3. The presence of uni-modal distribution shift leads to specific changes in the attention logit of two tokens from the shifted-modality (both tokens belonging to the single shifted modality). The variance of that change tends to be greater than that of two inter-modal tokens, i.e., one token from shifted modality and another token from the unshifted modality (We first prove the expectation of the change is zero in the appendix). This implies that the attention logits within the shifted modality have large fluctuations (e.g., high-variance inputs for the calculation of softmax of attention scores) after the shift occurs. As a result, the softmax outputs become more "peaked" within the shifted modalities' self-attention. Consequently, it becomes more likely that fewer cross-modal attention events occur, especially between the shifted and unshifted modalities. Ultimately, this has a negative impact on multi-modal fusion.

In addition to the above analysis, we also theoretically demonstrate that, under non-zero-mean shift, cross-modal fusion faces similar challenges from the expectation perspective, as detailed in the appendix.

### 3.4. Selective Adaptation with Model-specific Adapters and Router

Given that an unknown modality would be going through shift and harm multi-modal fusion, we propose to flexibly adapt the distribution shift of different modalities. We introduce two lightweight components on top of the base model: (I). Model-specific adapters: For the two modalities, we use two learnable matrices, $\Phi^{(v)}$ and $\Phi^{(a)}$, to fit the possible domain shift. (II). Router: Router is parameterized by modality shift semaphore $S = [s^{(v)}, s^{(a)}]$: A vector whose length equals the number of modalities, indicating the probabilities of a modality shift. In our case, the vector size is 2. During test-time, the two components for selective adaptation are updated end-to-end.

### 3.5. Forward Pass

**Feature Extraction** During test-time, the test samples arrive in batches $(\boldsymbol{x}^{(a)}, \boldsymbol{x}^{(v)}) \in \mathcal{B}$. Our model's forward process, illustrated in Fig. 3, first, extract features from the test data using the modality-specific encoders :

$$\begin{aligned} \boldsymbol{z}^{(a)} &= f^{(a)}(\boldsymbol{x}^{(a)}), \\ \boldsymbol{z}^{(v)} &= f^{(v)}(\boldsymbol{x}^{(v)}). \end{aligned} \quad (4)$$

**Selective Adaptation** Next, each of these features multiplies the corresponding modality-specific adapter to form the distribution-aligned features:

$$\begin{aligned} \hat{\boldsymbol{z}}^{(v)} &= \boldsymbol{z}^{(v)}(\Phi^{(v)} + I), \\ \hat{\boldsymbol{z}}^{(a)} &= \boldsymbol{z}^{(a)}(\Phi^{(a)} + I), \end{aligned} \quad (5)$$

where $I$ is an identity matrix make the adaptation works in a "residual" way. It allows the adapter $\Phi$ to build on the identity mapping and learn only the necessary increments.

We use the modality shift semaphore to decide whether to use the original or the adapted features. To provide a better exploration of the solution space with differentiable approximation to a categorical distribution, we input the modality shift semaphore $S = [s^{(v)}, s^{(a)}]$ into the Gumbel-Softmax function (Jang et al., 2017). Concretely, we first sample a Gumbel noise vector $\mathbf{g}$:

$$\mathbf{g} = -\log(-\log(\mathbf{u})), \quad (6)$$

where $\mathbf{u} \sim U(0, 1)$ and $\mathbf{u}$ has the same dimension as $S$. We then add this Gumbel noise to our input logits to obtain the noisy logits. We apply the softmax function to the semaphore to get the selection weights $\boldsymbol{w}$:

$$\boldsymbol{w} = \text{softmax}\big((S + \mathbf{g})/\tau\big), \quad (7)$$

where $\tau$ is a scaling temperature. To make the obtained weight decides whether to use the original or the adapted features, we obtain a convex combination of the original feature and the adapted feature. The weights $\boldsymbol{w}$ act as coefficients for the adaptation choice, determining how much of the corresponding modality's feature should be adapted:

$$\begin{aligned} \tilde{\boldsymbol{z}}^{(v)} &= w^{(v)} \cdot \boldsymbol{z}^{(v)} + (1 - w^{(v)}) \cdot \hat{\boldsymbol{z}}^{(v)}, \\ \tilde{\boldsymbol{z}}^{(a)} &= w^{(a)} \cdot \boldsymbol{z}^{(a)} + (1 - w^{(a)}) \cdot \hat{\boldsymbol{z}}^{(a)}. \end{aligned} \quad (8)$$

The choice here is implemented in a soft way to achieve sufficient training for both adapters. When $w^{(v)} \approx 1$, the output features $\tilde{\boldsymbol{z}}^{(v)}$ are more similar to the adapted feature $\hat{\boldsymbol{z}}^{(v)}$, while $w^{(a)} \approx 0$ causes $\tilde{\boldsymbol{z}}^{(a)}$ to be close to the original feature $\hat{\boldsymbol{z}}^{(a)}$.

**Multi-Modal Fusion** Subsequently, the features pass through an attention layer, which follows the well-known design from (Vaswani et al., 2017), before entering the classifier to yield the class prediction $\hat{\boldsymbol{p}}$:

$$\hat{\boldsymbol{p}} = \text{Softmax}\big(f^{(c)}(\text{mean}(f^{(m)}(\tilde{\boldsymbol{z}}^{(v)}, \tilde{\boldsymbol{z}}^{(a)})))\big), \quad (9)$$

where $\text{mean}(\cdot)$ calculates the mean features of all tokens output by the fusion layer.

**Traverse Inference Schema** Test-time adaptation, unlike the classic supervise training framework, must produce predictions for performance evaluation as soon as the data arrive. If the router has not fully converged in the early data stream, performance may be compromised. To address this issue, we propose a straightforward schema called traverse inference. In this approach, predictions for testing are generated through an ensemble of multiple forward passes, each activating one of the adapters. For instance, both forward processes for $(\hat{\boldsymbol{z}}^{(a)}, \boldsymbol{z}^{(v)})$ and $(\boldsymbol{z}^{(a)}, \hat{\boldsymbol{z}}^{(v)})$ are traversed. The final predictions given by choosing the most confident predictions from these outputs. It is worth noting that this schema is only for inference, adaptation only needs one forward routed by the selective adaptation schema.

### 3.6. Adaptation through Self-Training

To produce supervision signals without annotations, we use the self-training techniques that common in TTA and other relevant tasks (Sohn et al., 2020; Lee, 2013).

**Self-Training Loss** We first generate pseudo-labels based on the model's predictions $\hat{\boldsymbol{p}}$. The class index corresponding to the most confident prediction is used as the pseudo-label $\hat{p} = \arg\max_i(\hat{\boldsymbol{p}})_i$. Additionally, we threshold each sample's loss according to the maximum probability to ensure accurate self-supervision:

$$\mathcal{L}_{self} = \mathbb{1}(\max_i(\hat{\boldsymbol{p}})_i > \tau)\text{CE}(\hat{p}, \hat{\boldsymbol{p}}), \quad (10)$$

where threshold $\tau$ filters out less confident predictions. $\mathbb{1}(\cdot)$ is the indicator which transforms the bool value in it to 1 or 0. $\text{CE}(\cdot)$ is the cross-entropy loss function (The loss is averaged in each batch. We omit this for simplicity of illustration.).

**Balance Loss** Following (Yang et al., 2024; Zhou et al., 2023), we use the negative entropy loss term $\mathcal{L}_{bal}$ to regularize a balanced class distribution of the test batch. The details are in the appendix.

**Overall Self-Training Objective** The final adaptation objective combines these two losses:

$$\mathcal{L} = \mathcal{L}_{bal} + \alpha\mathcal{L}_{self}, \quad (11)$$

where $\alpha$ is a weight between the two terms. By minimizing this combined loss, the model can benefit from both the regularization of the class distribution via $L_{bal}$ and the exploitation of test data through pseudo-labeling in $L_{self}$.

## 4. Experiments

### 4.1. Experiment setting

**Dateset Construction** To validate the effectiveness of our method, we perform comparison experiments and on two multi-modal datasets, Kinetics50 (Kay et al., 2017; Peng et al., 2022) and VGGSound (Chen et al., 2020), with diverse domain shifts. The dataset construction and corruption follow the procedures in (Hendrycks & Dietterich, 2019; Yang et al., 2024). Specially, we use the Kinetics50 dataset, which is a 50-class subset of the Kinetics dataset and contains 29,204 YouTube videos of human actions (e.g., wrestling or eating cake), 15 types of video shifts are introduced. These include Gaussian noise, shot noise, impulse noise, defocus blur, glass blur, motion blur, zoom blur, snow, frost, fog, brightness, elastic, pixelate, contrast, and JPEG. Additionally, 6 types of audio corruptions-Gaussian noise, traffic noise, crowd noise, rainy noise, thunder noise, and windy noise-are applied. The videos in Kinetics50 are trimmed to a duration of 10 seconds as per (Yang et al., 2024). For the VGGSound dataset, which consists of 14,046

*Table 1.* Comparisons with SOTA methods on Kinetics50-C benchmark with corrupted video modality. "LF" refers to late fusion, "AF" refers to attention-based fusion. The results are the mean values among 5 random seeds, and the best results are highlighted in bold. Performance of other methods are from (Yang et al., 2024).

| Methods | | Noise | | | Blur | | | | Weather | | | | Digital | | | | Avg. |
|---|---|---|---|---|---|---|---|---|---|---|---|---|---|---|---|---|---|
| | | Gauss. | Shot | Impul. | Defoc. | Glass | Mot. | Zoom | Snow | Frost | Fog | Brit. | Contr. | Elas. | Pix. | JPEG | |
| LF | Source | 31.8 | 33.4 | 31.7 | 64.0 | 54.3 | 67.5 | 61.9 | 50.9 | 54.8 | 38.4 | 72.3 | 44.0 | 60.2 | 61.7 | 56.4 | 52.2 |
| | MM-TTA | 46.2 | 46.6 | 46.1 | 58.8 | 55.7 | 62.6 | 58.7 | 52.6 | 54.4 | 48.5 | 69.1 | 49.3 | 57.6 | 56.4 | 54.6 | 54.5 |
| | Tent | 28.6 | 29.8 | 28.3 | 63.4 | 51.1 | 67.7 | 61.7 | 46.5 | 51.3 | 24.5 | 72.3 | 38.6 | 60.7 | 61.8 | 54.9 | 49.4 |
| | ETA | 31.8 | 33.3 | 31.6 | 64.2 | 54.6 | 67.7 | 62.2 | 51.3 | 54.7 | 38.1 | 72.5 | 44.2 | 60.4 | 62.0 | 57.0 | 52.4 |
| | SAR | 31.9 | 33.3 | 31.7 | 63.8 | 54.0 | 67.7 | 61.8 | 50.7 | 54.5 | 38.8 | 72.3 | 44.0 | 60.3 | 62.0 | 56.5 | 52.2 |
| | READ | 34.0 | 34.5 | 33.8 | 65.3 | 57.7 | 68.7 | 64.9 | 56.1 | 57.5 | 41.1 | 73.2 | 48.7 | 62.9 | 64.6 | 59.2 | 54.8 |
| AF | Source | 46.8 | 48.0 | 46.9 | 67.5 | 62.2 | 70.8 | 66.7 | 61.6 | 60.3 | 46.7 | 75.2 | 52.1 | 65.7 | 66.5 | 61.9 | 59.9 |
| | Tent | 46.3 | 47.0 | 46.3 | 67.2 | 62.5 | 71.0 | 67.6 | 63.1 | 61.1 | 34.9 | 75.4 | 51.6 | 66.8 | 67.2 | 62.7 | 59.4 |
| | ETA | 46.8 | 47.6 | 47.1 | 67.2 | 62.7 | 70.6 | 67.2 | 62.3 | 60.9 | 46.7 | 75.2 | 52.4 | 65.9 | 66.8 | 62.5 | 60.1 |
| | SAR | 46.7 | 47.4 | 46.8 | 67.0 | 61.9 | 70.4 | 66.4 | 61.8 | 60.6 | 46.0 | 75.2 | 52.1 | 65.7 | 66.4 | 62.0 | 59.8 |
| | READ | 49.4 | 49.7 | 49.0 | 68.0 | 65.1 | **71.2** | **69.0** | 64.5 | 64.4 | 57.4 | **75.5** | 53.6 | 68.3 | 68.0 | 65.1 | 62.5 |
| | Ours | **52.6** | **52.3** | **52.0** | **68.7** | **68.0** | 70.7 | 68.8 | **65.2** | **66.6** | **64.3** | 74.6 | **57.4** | **70.5** | **69.0** | **66.2** | **64.5** |

10-second videos of everyday audio events labeled into 309 classes (such as snake rattling or pheasant crowing), the same uni-modal shifts as in Kinetics50 are constructed.

**Model and Training Protocol**  For fair comparisons, all compared methods use the same backbone models, pre-trained parameters and training protocol. The CAV-MAE (Gong et al., 2023) model is pretrained on the training sets of Kinetics50 and VGGSound dataset (To simulate the TTA setting, the TTA algorithm does not access training data. The original training set of Kinetics50 and VGGSound, as source domain, are only used for pre-training). The hyperparameters and other details are in the appendix.

**Compared Methods**  To comprehensively verify the effectiveness of our methods, We compare our method with four SOTA TTA methods, including the direct transfer of single modality TTA methods, namely Tent (Wang et al., 2020), ETA (Niu et al., 2022), and SAR (Niu et al., 2023), on the multi-modal TTA setting and the multi-modal TTA methods: MM-TTA (Shin et al., 2022) and READ (Yang et al., 2024). Following (Yang et al., 2024), we report the these methods' results with two versions of modality fusion: Late fusion (LF), e.g., directly emsembling the outputs of modality-specific encoders as the input to the classification head. Attention-based Fusion (AF), e.g., the outputs of the modality-specific encoders are first fused through an attention layer, and the resulting fused representation is then fed into the classification head.

### 4.2. Experimental evaluation

**Q1: How is the performance of our method compared with SOTA TTA method?**

The presented tables 1, 2, and 3, compare our method with SOTA techniques across different datasets (Kinetics50-C and VGGSound-C) and modality shift types (corrupted video and audio modalities). In all three tables, for vari-

ous corruption types such as noise, blur, weather, and digital in the video modality, and noise and weather in the audio modality, our method achieves the best average results. In the Kinetics50-C benchmark with corrupted video modality (Table 1), across all corruption types like Gaussian, shot, and impulse noise, defocus, glass, motion, zoom blur, snow, frost, fog, brightness, contrast, elastic, pixelate, and JPEG compression, our method has the highest average accuracy (64.5). Similarly, in the audio-corrupted Kinetics50-C benchmark (left part of Table 3), for noise (Gaussian, traffic, crowd) and weather (rain, thunder, wind) corruptions, our method attains the best average accuracy (71.5). In the VGGSound-C benchmark, whether with corrupted video (Table 2) or audio (right part of Table 3) modalities, our method leads in terms of average performance.

For the Kinetics50-C dataset, video can be considered a dominant modality. In the case of video-related corruptions (Table 1), the performance gap between our method and others is more pronounced. This indicates a more substantial gain in the dominant modality shift scenario. On the other hand, for the VGGSound-C dataset with video shift (Table 2), the performance improvement of our method over SOTA is relatively less. This shows that on VGGSound-C with vide shift, there is less room for performance enhancement compared to the dominant modality shift cases like video in Kinetics50-C.

In summary, the experimental results clearly demonstrate that our method outperforms SOTA across different modality shift types and datasets, with a more significant gain in dominant modality shifts, while also highlighting the performance limitations in certain datasets and modality-corruption combinations.

**Q2: How is the shifted data behaves from the attention view?**

In our experiments as shown Fig. 4, we analyze the atten-

Table 2. Comparisons with SOTA methods on VGGSound-C benchmark with corrupted video modality.

| | Methods | Noise | | | Blur | | | | Weather | | | | Digital | | | | Avg. |
|---|---|---|---|---|---|---|---|---|---|---|---|---|---|---|---|---|---|
| | | Gauss. | Shot | Impul. | Defoc. | Glass | Mot. | Zoom | Snow | Frost | Fog | Brit. | Contr. | Elas. | Pix. | JPEG | |
| LF | Source | 37.7 | 36.5 | 37.8 | 52.7 | 51.3 | 55.2 | 53.7 | 51.9 | 52.3 | 50.4 | 55.3 | 45.2 | 52.5 | 51.7 | 52.3 | 49.1 |
| | MM-TTA | 7.1 | 7.3 | 7.3 | 44.8 | 41.5 | 48.0 | 45.5 | 27.4 | 23.5 | 30.5 | 46.9 | 24.2 | 40.3 | 40.7 | 45.7 | 32.0 |
| | Tent | 7.6 | 6.8 | 7.2 | 53.1 | 52.1 | 55.5 | 54.5 | 52.6 | 32.7 | 16.0 | 55.9 | 16.6 | 52.6 | 54.2 | 53.1 | 38.0 |
| | ETA | 37.7 | 36.5 | 37.7 | 53.2 | 52.3 | 56.0 | 54.4 | 52.4 | 52.9 | 51.0 | 55.0 | 45.2 | 53.5 | 52.3 | 52.7 | 49.5 |
| | SAR | 37.7 | 36.4 | 37.7 | 52.8 | 51.5 | 55.5 | 53.9 | 51.9 | 52.5 | 50.4 | 55.4 | 44.8 | 52.7 | 51.8 | 52.3 | 49.2 |
| | READ | 42.1 | 41.5 | 42.1 | 49.3 | 50.9 | 53.5 | 52.5 | 50.6 | 52.1 | 51.1 | 54.0 | 46.2 | 52.5 | 49.1 | 50.2 | 49.2 |
| AF | Source | 52.8 | 52.7 | 52.7 | 57.2 | 57.2 | 58.7 | 57.6 | 56.4 | 56.6 | 55.6 | 58.9 | 53.7 | 56.9 | 55.8 | 56.9 | 56.0 |
| | Tent | 52.7 | 52.7 | 52.7 | 56.7 | 56.5 | 57.9 | 57.2 | 55.9 | 56.3 | 56.3 | 58.4 | 54.0 | 57.4 | 56.2 | 56.7 | 55.8 |
| | ETA | 53.0 | 52.8 | 53.0 | 57.2 | 57.1 | 58.6 | 57.8 | 56.3 | 56.8 | 56.4 | 59.0 | 54.1 | 57.4 | 56.1 | 57.0 | 56.2 |
| | SAR | 52.9 | 52.8 | 52.9 | 57.2 | 57.1 | 58.6 | 57.6 | 56.3 | 56.7 | 55.9 | 58.9 | 54.0 | 57.0 | 56.0 | 57.0 | 56.1 |
| | READ | 53.6 | 53.6 | 53.5 | **57.9** | 57.7 | **59.4** | **58.8** | **57.2** | 57.8 | **55.0** | **59.9** | 55.2 | **58.6** | 57.1 | **57.9** | **56.9** |
| | Ours | **53.9** | **53.9** | **54.0** | 57.6 | **58.0** | 59.0 | 58.6 | 56.9 | 57.0 | **56.6** | 59.8 | 54.7 | **58.6** | 56.7 | **57.9** | **56.9** |

Table 3. Comparisons with SOTA methods on Kinetics50-C (left) and VGGSound-C (right) benchmarks with corrupted audio modality.

| | Methods | Kinetics50-C | | | | | | | VGGSound-C | | | | | | |
|---|---|---|---|---|---|---|---|---|---|---|---|---|---|---|---|
| | | Noise | | | Weather | | | Avg. | Noise | | | Weather | | | Avg. |
| | | Gauss. | Traff. | Crowd. | Rain | Thund. | Wind | | Gauss. | Traff. | Crowd. | Rain | Thund. | Wind | |
| LF | Source | 71.1 | 67.8 | 67.4 | 67.4 | 70.6 | 68.6 | 68.8 | 29.5 | 17.1 | 22.6 | 17.3 | 33.7 | 20.6 | 23.5 |
| | MM-TTA | 70.8 | 69.2 | 68.5 | 69.0 | 69.8 | 69.4 | 69.4 | 14.1 | 5.2 | 6.4 | 6.9 | 8.6 | 4.5 | 7.6 |
| | Tent | 71.1 | 68.6 | 67.8 | 67.4 | 71.2 | 68.9 | 69.2 | 6.4 | 2.1 | 2.9 | 1.9 | 9.5 | 3.1 | 4.3 |
| | ETA | 71.2 | 67.9 | 67.5 | 67.8 | 70.9 | 68.7 | 69.0 | 28.8 | 17.1 | 22.4 | 17.4 | 33.8 | 20.4 | 23.3 |
| | SAR | 71.1 | 67.5 | 67.4 | 67.4 | 70.6 | 68.6 | 68.8 | 28.5 | 16.6 | 22.4 | 17.4 | 33.7 | 20.2 | 23.1 |
| | READ | 71.3 | 68.5 | 68.5 | 68.4 | 71.8 | 69.0 | 69.6 | 36.4 | 25.3 | 28.9 | 27.3 | 35.6 | 26.6 | 30.0 |
| AF | Source | 73.7 | 65.5 | 67.9 | 70.3 | 67.9 | 70.3 | 69.3 | 37.0 | 25.5 | 16.8 | 21.6 | 27.3 | 25.5 | 25.6 |
| | Tent | 73.9 | 67.4 | 69.2 | 70.4 | 66.5 | 70.5 | 69.6 | 10.6 | 2.6 | 1.8 | 2.8 | 5.3 | 4.1 | 4.5 |
| | ETA | 73.7 | 66.1 | 68.5 | 70.3 | 67.9 | 70.1 | 69.4 | 39.2 | 26.1 | 22.9 | 26.0 | 31.7 | 30.4 | 29.4 |
| | SAR | 73.7 | 65.4 | 68.2 | 69.9 | 67.2 | 70.2 | 69.1 | 37.4 | 9.5 | 11.0 | 12.1 | 26.8 | 23.7 | 20.1 |
| | READ | 74.1 | 69.0 | 69.7 | 71.1 | 71.8 | 70.7 | 71.1 | 40.4 | 28.9 | 26.6 | 30.9 | 36.7 | 30.6 | 32.4 |
| | Ours | **74.5** | **69.6** | **70.5** | **71.4** | **72.0** | **71.0** | **71.5** | **41.5** | **31.8** | **30.9** | **32.6** | **38.9** | **32.6** | **34.7** |

tion mechanisms of our proposed method in the context of multi-modal fusion, particularly when dealing with corrupted video inputs. Shifted video features would exhibit smaller similarity to the unshifted audio modality. This discrepancy led to reduced attention allocation to the audio modality, ultimately resulting in poorer multi-modal fusion outcomes. When examining the attention maps generated by the pre-trained model, in the first row of Fig. 4, we note that there was a reduction in video-to-audio attention, as evidenced by the darker color in the top right of the first row of figures. In contrast, our method demonstrated a recovery in attention dynamics: the video modality not only attended to itself but also engaged more effectively with the audio modality, as highlighted by the lighter color in the top right of corresponding figures of the second row.

Furthermore, our method consistently achieved smaller variance across all five video shifts as shown in the number in each figure in Fig. 4, indicating a more stable and reliable attention pattern. These results suggest that our approach not only enhances the attention allocation towards relevant modalities but also contributes to a more coherent and consistent fusion of information across modalities.

Table 4. The comparison of number of trainable parameters. *Tent's computation cost represents a series of methods that tunes the parameters normalization layer.

| Methods | # of Params (million) | Time cost (seconds per epoch) | |
|---|---|---|---|
| | | Kinetics50-C | VGGSound-C |
| Tent* | 0.226 | 54.56 | OOM |
| READ | 1.772 | 35.43 | 190.70 |
| Ours | 1.180 | 35.96 | 191.76 |

## Q3: How is the efficiency of our method compared with other TTA method?

Test-time training methods prioritize computational efficiency as they aim to optimize model performance without incurring excessive resource costs, particularly in environments with limited GPU memory or time constraints. In this context, we evaluated various approaches in Table 4, including Tent, READ, and our proposed method. Tent, which adjusts normalization layer parameters, boasts the fewest trainable parameters at 0.226 million. However, it significantly increases computational demands, leading to out-of-memory (OOM) errors on the VGGSound-C dataset. This inefficiency stems from the necessity to save gradient

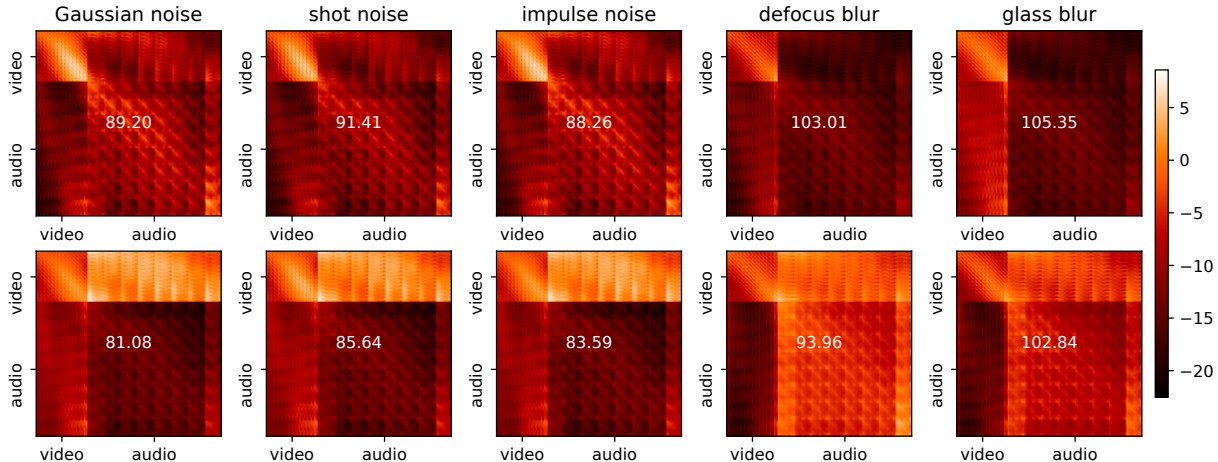

*Figure 4.* Attention map (logit) of the multi-modal fusion layer on Kinetics50-C dataset with five video shifts. The first row shows the pre-trained model's attention maps, the second row is our method's attention maps. The number in the middle is the average variance over the whole dataset. The first 196 tokens are from the video modality, the last 512 tokens are from the audio modality. The lighter the color, the bigger the attention logit as shown in the right color bar.

graphs for shallow layers, which complicates the training process and escalates memory usage. In contrast, both our method and READ, despite having more parameters—1.180 million and 1.772 million respectively—demonstrate greater computational efficiency. They require less GPU memory and avoid OOM issues, highlighting a critical finding that a higher parameter count does not inherently compromise resource efficiency. This analysis underscores the importance of balancing parameter count and computational demands in the design of effective test-time training methods. The detailed hardware information is provided the appendix.

### Q4: Are the components of our method effective?

In our ablation study, we systematically strip off three components one by one from our proposed method to evaluate their contributions to performance. We first try to change the Gumbel-softmax to direct softmax function. Then, we removed the test-time ensemble schema, followed by the selective adaptation schema, which utilizes the adapted features across all modalities. Without these adapters, the model reverts to the pre-trained source model. The results, summarized in Table 5, demonstrate the effectiveness of the components in our method.

Two plots in Fig. 5 show the sensitivity test of two hyperparameters. The first plot shows that as the batch size increases from 8 to 128, the accuracy for both Gaussian and shot noise declines. This indicates that the batch size has a strong influence on the model's performance. Even though layer normalization is used, the significant impact of the batch size implies that the update rate of our model, which is related to the batch size, is crucial in the TTA task. The second plot shows that our method is not sensitive to the temperature for Gumbel-softmax. We show more sensitivity

*Table 5.* Ablation study on Kinetics50-C with five video shifts.

| Ablation | Gauss. | Traff. | Crowd. | Rain | Thund. |
|---|---|---|---|---|---|
| Ours | **52.6** | **52.3** | **52.0** | **68.7** | **68.0** |
| -Gumbel-softmax | 52.0 | 51.9 | 51.4 | 67.8 | **68.0** |
| -Traverse inference | 51.5 | 51.5 | 51.1 | 67.2 | 67.5 |
| -Selective adaptation | 50.3 | 50.7 | 50.5 | 65.1 | 65.7 |
| Source model | 46.8 | 48.0 | 46.9 | 67.5 | 62.2 |

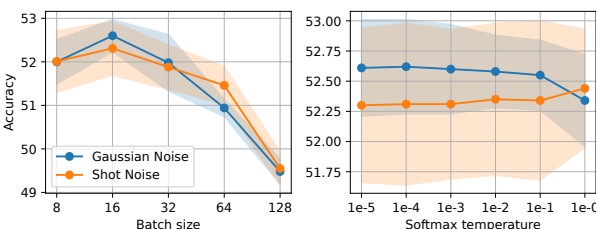

*Figure 5.* Sensitivity test of the loss coefficient and batch size.

tests in the appendix.

## 5. Conclusion

In this paper, we systematically define and study the problem of test-time uni-modal distribution shift. We find that previous methods do not efficiently transfer to this new setting, primarily due to a negative transfer phenomenon. Without knowledge of which modality will experience a distribution shift, blindly applying adaptation methods to modality-specific modules is not only redundant but also potentially has a negative effect. Furthermore, we theoretically analyze how uni-modal distribution shift hinders a stable attention mechanism in multi-modal fusion. To address the unique challenges posed by this novel problem, we propose

a simple method with the core idea of selective adaptation. The model maintains a router to choose which modality to adapt and modality-specific adapters in a self-training fashion. As a result, our method effectively achieves test-time adaptation for uni-modal distribution shift. We conduct extensive experiments to demonstrate the superiority of our method. From the perspectives of attention maps and computational cost, we also prove the practical significance of our method. We anticipate that the proposed new problem has both application and research value. By addressing these challenges, the broader implications of our work can be further expanded, leading to more reliable and deployable multi-modal models.

## Acknowledgements

This work was supported by the National Natural Science Foundation of China under Grants 62325206, the Key Research and Development Program of Jiangsu Province under Grant BE2023016-4, Natural Science Research Start-up Foundation of Recruiting Talents of Nanjing University of Posts and Telecommunications (Grant No. NY224061, NY224103), and National Natural Science Foundation of China (Grant No. 624B2100).

## Impact Statement

This paper presents work whose goal is to advance the field of Machine Learning. There are many potential societal consequences of our work, none of which we feel must be specifically highlighted here.

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

## A. Theoretical Analysis

### A.1. Analysis of Cross-Modal Attention under Uni-Modal Distribution Shift with Zero-Mean Additive Noise

The self-attention mechanism lies at the core of multi-modal fusion. This mechanism is fundamental as it allows the model to selectively focus on different regions within and across modalities. It aggregates intra-modal and inter-modal token representations with distinct weights, following the formula:

$$\text{Attention}(Q(\boldsymbol{z}), K(\boldsymbol{z}), V(\boldsymbol{z})) = \text{softmax}\left(\frac{Q(\boldsymbol{z})K(\boldsymbol{z})^T}{\sqrt{d_k}}\right) V(\boldsymbol{z}), \tag{12}$$

where query $Q(\cdot)$, key $K(\cdot)$ and value $V(\cdot)$ are the linear transform on the tokens' representation $\boldsymbol{z}$, $d_k$ is the scale factor. The attention logit, which determines the relative importance of token pairs in the self-attention mechanism, is defined as the inner product between the query and key matrices prior to the softmax operation. Its value varies depending on the representations of a pair of tokens. Formally, we define the *attention logit* in Definition 3.1.

**Definition A.1** (Attention Logit). The *attention logit* (AL) is defined as the inner product between query and key matrices:

$$\text{AL}(\boldsymbol{z}_i, \boldsymbol{z}_j) = Q(\boldsymbol{z}_i)K(\boldsymbol{z}_j)^T = \boldsymbol{z}_i\left(W^Q W^{K^T}\right)\boldsymbol{z}_j^T, \tag{13}$$

where $\boldsymbol{z}_i \in \mathbb{R}^F$ and $\boldsymbol{z}_j \in \mathbb{R}^F$ are a pair of embedding of tokens involved in the self-attention calculation, and the linear projection matrices $W^Q, W^K \in \mathbb{R}^{F \times F}$ are considered to be constant.

However, the presence of a distribution shift in any modality can have adverse effects on this self-attention mechanism. Without loss of generality, we analyze the case of a shifted audio modality. Under the uni-modal distribution shift, tokens from the shifted modality are corrupted. We denote the corrupted tokens as $\acute{\boldsymbol{z}}_i^{(a)}$ and $\acute{\boldsymbol{z}}_j^{(a)}$, and the tokens from the unshifted modality as $\boldsymbol{z}_k^{(v)}$.

Following the work of distribution shift with additive noise (Kim et al., 2020), we assume that, after the distribution shift, zero-mean noise $\varepsilon$ is added to the tokens $z$ of the input modality (Song et al., 2015; Huang et al., 2022; Yasarla et al., 2024), Specifically,

$$\acute{\boldsymbol{z}} = \boldsymbol{z} + \varepsilon \quad \text{where} \quad \mathbb{E}[\varepsilon] = \mathbf{0}_{1 \times n}, \tag{14}$$

The noise $\varepsilon$ is independent of the distribution of the clean token $\boldsymbol{z}$ and is independent across different tokens.

Let $\boldsymbol{z}_i^{(a)}$ and $\boldsymbol{z}_k^{(v)}$ represent tokens from different modalities, where $\boldsymbol{z}_i^{(a)}$ is corrupted to $\acute{\boldsymbol{z}}_i^{(a)}$ by $\varepsilon_i$. We can show that the expectation of the attention logit remains unchanged after the corruption:

$$\mathbb{E}\left[Q(\acute{\boldsymbol{z}}_i^{(a)})K(\boldsymbol{z}_k^{(v)})^T - Q(\boldsymbol{z}_i^{(a)})K(\boldsymbol{z}_k^{(v)})^T\right] \tag{15}$$

$$= \mathbb{E}\left[(\boldsymbol{z}_i^{(a)} + \varepsilon_i)W^Q W^{K^T}(\boldsymbol{z}_k^{(v)})^T - \boldsymbol{z}_i^{(a)}W^Q W^{K^T}(\boldsymbol{z}_k^{(v)})^T\right] \tag{16}$$

$$= \mathbb{E}\left[\varepsilon W^Q W^{K^T}(\boldsymbol{z}_k^{(v)})^T\right] \tag{17}$$

$$= \left\langle\mathbb{E}[\varepsilon_i], \mathbb{E}\left[W^Q W^{K^T}(\boldsymbol{z}_i^{(a)})^T\right]\right\rangle \tag{18}$$

$$= 0 \tag{19}$$

Let $\boldsymbol{z}_i^{(a)}$ and $\boldsymbol{z}_j^{(a)}$ represent tokens from the same modality. The expectation of the attention logit also remains unchanged.

$$\mathbb{E}\left[Q(\acute{\boldsymbol{z}}_i^{(a)})K(\acute{\boldsymbol{z}}_j^{(a)})^T - Q(\boldsymbol{z}_i^{(a)})K(\boldsymbol{z}_j^{(a)})^T\right] \tag{20}$$

$$= \mathbb{E}\left[(\boldsymbol{z}_i^{(a)} + \varepsilon_i)W^Q W^{K^T}(\boldsymbol{z}_j^{(a)} + \varepsilon_j)^T - \boldsymbol{z}_i^{(a)}W^Q W^{K^T}(\boldsymbol{z}_j^{(a)})^T\right] \tag{21}$$

$$= \mathbb{E}\left[\varepsilon_i W^Q W^{K^T}(\boldsymbol{z}_j^{(a)})^T + \boldsymbol{z}_i^{(a)}W^Q W^{K^T}\varepsilon_j^T + \varepsilon_i W^Q W^{K^T}\varepsilon_j^T\right] \tag{22}$$

$$= \left\langle\mathbb{E}[\varepsilon_i], \mathbb{E}\left[W^Q W^{K^T}(\boldsymbol{z}_j^{(a)})^T\right]\right\rangle + \left\langle\mathbb{E}\left[\boldsymbol{z}_i^{(a)}W^Q W^{K^T}\right], \mathbb{E}[\varepsilon_j]\right\rangle + \left\langle\mathbb{E}[\varepsilon_i], \mathbb{E}\left[W^Q W^{K^T}\varepsilon_j\right]\right\rangle \tag{23}$$

$$= 0 \tag{24}$$

Since the expectation of the attention logit of the noisy token is the same as that of the clean token, we then turn our attention to the variance.

The variance of the difference between the attention logit of tokens from different modalities:

$$\mathbb{D}\Big[Q(\acute{z}_i^{(a)})K(z_k^{(v)})^T - Q(z_i^{(a)})K(z_k^{(v)})^T\Big] \tag{25}$$

$$= \mathbb{D}\big[\varepsilon_i W^Q W^{K^T}(z_k^{(v)})^T\big] \tag{26}$$

$$= \mathbb{E}\big[\big(\varepsilon_i W^Q W^{K^T}(z_k^{(v)})^T\big)^2\big] - \big(\mathbb{E}[\varepsilon_i W^Q W^{K^T}(z_k^{(v)})^T]\big)^2 \tag{27}$$

$$= \Big\|\mathbb{E}\big[\big(\varepsilon_i W^Q W^{K^T}(z_k^{(v)})^T\big)^2\big]\Big\|_2 \tag{28}$$

$$= \Big\|\mathbb{E}[\varepsilon_i W^Q W^{K^T}(z_k^{(v)})^T \varepsilon_i W^Q W^{K^T}(z_k^{(v)})^T]\Big\|_2 \tag{29}$$

$$\leq \sigma_{max}^2 \Big\|\mathbb{E}[\varepsilon_i (z_k^{(v)})^T \varepsilon_i (z_k^{(v)})^T]\Big\|_2 \tag{30}$$

$$= \sigma_{max}^2 \Big\|\mathbb{E}\big[z_k^{(v)} \varepsilon_i^T \varepsilon_i (z_k^{(v)})^T\big]\Big\|_2 \tag{31}$$

$$= \sigma_{max}^2 \Big\|\mathbb{E}\big[\mathrm{tr}\big(z_k^{(v)} \varepsilon_i^T \varepsilon_i (z_k^{(v)})^T\big)\big]\Big\|_2 \tag{32}$$

$$= \sigma_{max}^2 \Big\|\mathbb{E}\big[\mathrm{tr}\big(\varepsilon_i^T \varepsilon_i (z_k^{(v)})^T z_k^{(v)}\big)\big]\Big\|_2 \tag{33}$$

$$= \sigma_{max}^2 \Big\|\mathrm{tr}\big(\mathbb{E}[\varepsilon_i^T \varepsilon_i] \mathbb{E}[(z_k^{(v)})^T z_k^{(v)}]\big)\Big\|_2 \tag{34}$$

$$\leq \sigma_{max}^2 \Big\|\mathrm{tr}\big(\mathbb{E}[\varepsilon_i^T \varepsilon_i]\big) \mathrm{tr}\big(\mathbb{E}[(z_k^{(v)})^T z_k^{(v)}]\big)\Big\|_2 \tag{35}$$

$$= \sigma_{max}^2 \Big\|\mathrm{tr}(\mathbb{E}[\varepsilon_i \varepsilon_i^T]) \mathrm{tr}(\mathbb{E}[z_k^{(v)} (z_k^{(v)})^T])\Big\|_2 \tag{36}$$

$$= \sigma_{max}^2 \|\mathbb{E}[\varepsilon_i \varepsilon_i^T]\|_2 \cdot \|\mathbb{E}[z_k^{(v)} (z_k^{(v)})^T]\|_2 \tag{37}$$

$$= \sigma_{max}^2 \mathbb{E}[\varepsilon_i \varepsilon_i^T] \cdot \mathbb{E}[z_k^{(v)} (z_k^{(v)})^T] \tag{38}$$

$$= \sigma_{max}^2 \mathbb{E}[\|\varepsilon_i\|_2^2] \cdot \mathbb{E}[\|z_k^{(v)}\|_2^2] \tag{39}$$

The combination of linear transformation $W^Q W^{K^T}$ can be decomposed by singular value decomposition into $U\Sigma V^T$, where $\sigma_{max} > 0$ represents the largest singular value of $\Sigma$ (excluding the trivial case of $W^Q W^{K^T}$ is all-zero matrix). The inequality $\|aU\Sigma V^T\|_2 <= \sigma_{max}\|a\|_2$ is used to obtain the upper-bound in Eq. 30. $\varepsilon_i^T \varepsilon_i$ denotes a positive semidefinite outer product and $\mathrm{tr}(\cdot)$ is the trace. The inequality $0 \leq \mathrm{tr}(AB) \leq \mathrm{tr}(A)\mathrm{tr}(B)$ for positive semidefinite matrices is used to deduce Eq. 35. We denote Eq. 39 as $\sup\big(\mathbb{D}\Big[Q(\acute{z}_i^{(a)})K(z_k^{(v)})^T - Q(z_i^{(a)})K(z_k^{(v)})^T\Big]\big)$.

Similarly, for the variance of the attention logit bias between tokens from the same modality:

$$\mathbb{D}\Big[Q(\acute{z}_i^{(a)})K(\acute{z}_j^{(a)})^T - Q(z_i^{(a)})K(z_j^{(a)})^T\Big] \tag{40}$$

$$= \mathbb{D}\Big[\varepsilon_i W^Q W^{K^T}(z_j^{(a)})^T + z_i^{(a)} W^Q W^{K^T} \varepsilon_j^T + \varepsilon_i W^Q W^{K^T} \varepsilon_j^T\Big] \tag{41}$$

$$= \mathbb{D}\Big[\varepsilon_i W^Q W^{K^T}(z_j^{(a)})^T\Big] + \mathbb{D}\Big[z_i^{(a)} W^Q W^{K^T} \varepsilon_j\Big] + \mathbb{D}\Big[\varepsilon_i W^Q W^{K^T} \varepsilon_j^T\Big] \tag{42}$$

$$+ 2\mathrm{Cov}\Big[\varepsilon_i W^Q W^{K^T}(z_j^{(a)})^T, z_i^{(a)} W^Q W^{K^T} \varepsilon_j^T\Big] + 2\mathrm{Cov}\Big[\varepsilon_i W^Q W^{K^T}(z_i^{(a)})^T, \varepsilon_i W^Q W^{K^T} \varepsilon_j^T\Big] \tag{43}$$

$$+ 2\mathrm{Cov}\Big[z_i^{(a)} W^Q W^{K^T} \varepsilon_j, \varepsilon_i W^Q W^{K^T} \varepsilon_j^T\Big] \tag{44}$$

$$= \mathbb{D}\Big[\varepsilon_i W^Q W^{K^T}(z_j^{(a)})^T\Big] + \mathbb{D}\Big[z_i^{(a)} W^Q W^{K^T} \varepsilon_j\Big] + \mathbb{D}\Big[\varepsilon_i W^Q W^{K^T} \varepsilon_j^T\Big] \tag{45}$$

$$= \mathbb{E}\big[(\varepsilon_i W^Q W^{K^T}(z_j^{(a)})^T)^2\big] + \mathbb{E}\big[(z_i^{(a)} W^Q W^{K^T} \varepsilon_j^T)^2\big] + \mathbb{E}\big[(\varepsilon_i W^Q W^{K^T} \varepsilon_j^T)^2\big] \tag{46}$$

$$\leq \sigma_{max}^2 \|\mathbb{E}[\varepsilon_i \varepsilon_i^T]\|_2 \cdot \|\mathbb{E}[z_j^{(a)} (z_j^{(a)})^T]\|_2 + \sigma_{max}^2 \|\mathbb{E}[\varepsilon_j \varepsilon_j^T]\|_2 \cdot \|\mathbb{E}[z_i^{(a)} (z_i^{(a)})^T]\|_2 + \sigma_{max}^2 \|\mathbb{E}[\varepsilon_i \varepsilon_i^T]\|_2 \cdot \|\mathbb{E}[\varepsilon_j \varepsilon_j^T]\|_2 \tag{47}$$

$$= \sigma_{max}^2 \|\mathbb{E}[\varepsilon_i \varepsilon_i^T]\|_2 \cdot \|\mathbb{E}[z_j^{(a)} (z_j^{(a)})^T]\|_2 + \sigma_{max}^2 \|\mathbb{E}[\varepsilon_i \varepsilon_i^T]\|_2 \cdot \|\mathbb{E}[z_i^{(a)} (z_i^{(a)})^T]\|_2 + \sigma_{max}^2 \|\mathbb{E}[\varepsilon_i \varepsilon_i^T]\|_2 \cdot \|\mathbb{E}[\varepsilon_j \varepsilon_j^T]\|_2 \tag{48}$$

$$= \sigma_{max}^2 \mathbb{E}[\varepsilon_i \varepsilon_i^T] \cdot \mathbb{E}[\boldsymbol{z}_j^{(a)}(\boldsymbol{z}_j^{(a)})^T] + \sigma_{max}^2 \mathbb{E}[\varepsilon_i \varepsilon_i^T] \cdot \mathbb{E}[\boldsymbol{z}_i^{(a)}(\boldsymbol{z}_i^{(a)})^T] + \sigma_{max}^2 \mathbb{E}[\varepsilon_i \varepsilon_i^T] \cdot \mathbb{E}[\varepsilon_j \varepsilon_j^T] \tag{49}$$

$$= \sigma_{max}^2 \mathbb{E}[\|\varepsilon_i\|_2^2] \cdot \mathbb{E}[\|\boldsymbol{z}_j^{(a)}\|_2^2] + \sigma_{max}^2 \mathbb{E}[\|\varepsilon_i\|_2^2] \cdot \mathbb{E}[\|\boldsymbol{z}_i^{(a)}\|_2^2] + \sigma_{max}^2 \mathbb{E}[\|\varepsilon_i\|_2^2] \cdot \mathbb{E}[\|\varepsilon_j\|_2^2] \tag{50}$$

Denote Eq. 50 as $\sup\left(\mathbb{D}\left[Q(\acute{\boldsymbol{z}}_i^{(a)})K(\acute{\boldsymbol{z}}_j^{(a)})^T - Q(\boldsymbol{z}_i^{(a)})K(\boldsymbol{z}_j^{(a)})^T\right]\right)$. We have

$$\sup\left(\mathbb{D}\left[Q(\acute{\boldsymbol{z}}_i^{(a)})K(\boldsymbol{z}_k^{(v)})^T - Q(\boldsymbol{z}_i^{(a)})K(\boldsymbol{z}_k^{(v)})^T\right]\right) - \sup\left(\mathbb{D}\left[Q(\acute{\boldsymbol{z}}_i^{(a)})K(\acute{\boldsymbol{z}}_j^{(a)})^T - Q(\boldsymbol{z}_i^{(a)})K(\boldsymbol{z}_j^{(a)})^T\right]\right) \tag{51}$$

$$= \sigma_{max}^2 \mathbb{E}[\|\varepsilon_i\|_2^2]\left(\mathbb{E}[\|\boldsymbol{z}_j^{(a)}\|_2^2] + \mathbb{E}[\|\boldsymbol{z}_i^{(a)}\|_2^2] + \mathbb{E}[\|\varepsilon_j\|_2^2] - \|\boldsymbol{z}_k^{(v)}\|_2^2\right) \tag{52}$$

We then state the following proposition:

**Proposition A.2.** *Under the additive shift assumption (Kim et al., 2020):*

$$\sup\left(\mathbb{D}\left[\underbrace{\mathrm{AL}(\acute{\boldsymbol{z}}_i^{(a)}, \boldsymbol{z}_k^{(v)}) - \mathrm{AL}(\boldsymbol{z}_i^{(a)}, \boldsymbol{z}_k^{(v)})}_{\text{change of cross-modal AL after shift}}\right]\right) < \sup\left(\mathbb{D}\left[\underbrace{\mathrm{AL}(\acute{\boldsymbol{z}}_i^{(a)}, \acute{\boldsymbol{z}}_j^{(a)}) - \mathrm{AL}(\boldsymbol{z}_i^{(a)}, \boldsymbol{z}_j^{(a)})}_{\text{change of intra-shited-modal AL after shift}}\right]\right) \tag{53}$$

$$\text{when } \mathbb{E}[\|\boldsymbol{z}_k^{(v)}\|_2^2] < \mathbb{E}[\|\boldsymbol{z}_i^{(a)}\|_2^2] + \mathbb{E}[\|\boldsymbol{z}_j^{(a)}\|_2^2] + \mathbb{E}[\|\varepsilon_j\|_2^2]$$

*where $\sup(\cdot)$ is the least upper bound. $\varepsilon_j$ is the noise related to token $\boldsymbol{z}_j^{(a)}$. $\mathbb{E}[\|\boldsymbol{z}_k^{(v)}\|_2^2]$, $\mathbb{E}[\|\boldsymbol{z}_i^{(a)}\|_2^2]$, $\mathbb{E}[\|\boldsymbol{z}_j^{(a)}\|_2^2]$, $\mathbb{E}[\|\varepsilon_j\|_2^2]$ are the expected value of the squared norm of the token representations from the different modality and noise, respectively.*

In practical terms, the expected value of the squared norm of the token representations gives us a measure of the average "energy" or magnitude of the vector across its possible realizations.

## A.2. Expectation Analysis of Cross-Modal Attention under Uni-Modal Disdistribution Shift with Non-Zero-Mean Additive Noise

In this section, we further analysis the case with non-zero-mean noise. The expectation of the attention logit between uncorrupted tokens as follow.

$$\mathbb{E}\left[Q(\boldsymbol{z}_i^{(a)})K(\boldsymbol{z}_j^{(a)})^T\right] = \left\langle \mathbb{E}[\boldsymbol{z}_i^{(a)}W^Q], \mathbb{E}[\boldsymbol{z}_j^{(a)}W^K]\right\rangle + \mathrm{tr}(\Sigma_{(\boldsymbol{z}_i^{(a)}, \boldsymbol{z}_j^{(a)})}) \tag{54}$$

$$= \mathbb{E}[\boldsymbol{z}_i^{(a)}]W^Q W^{K^T}(\mathbb{E}[(\boldsymbol{z}_j^{(a)})])^T + \mathrm{tr}(\Sigma_{(\boldsymbol{z}_i^{(a)}, \boldsymbol{z}_j^{(a)})}) \tag{55}$$

The matrix $\Sigma_{(\boldsymbol{z}_i^{(a)}, \boldsymbol{z}_j^{(a)})}$ denotes the covariance between $\boldsymbol{z}_i^{(a)}W^Q$ and $\boldsymbol{z}_j^{(a)}W^K$. Similarly, we have additive noise as:

$$\acute{\boldsymbol{z}}_i^{(a)} = \boldsymbol{z}_i^{(a)} + \varepsilon_i, \; \acute{\boldsymbol{z}}_j^{(a)} = \boldsymbol{z}_j^{(a)} + \varepsilon_j \tag{56}$$

where the noise $\varepsilon$ is independent of tokens' feature $\boldsymbol{z}$ and follows an independent and identically-distributed (i.i.d.) distribution across different tokens. Unlike Sec. A.1, its mean is not zero. The expectation of the attention logit between the clean token and corrupted token is as follows:

$$\mathbb{E}\left[Q(\acute{\boldsymbol{z}}_i^{(a)})K(\boldsymbol{z}_k^{(v)})^T\right] = \mathbb{E}\left[Q(\boldsymbol{z}_i^{(a)} + \varepsilon_i)K(\boldsymbol{z}_k^{(v)})^T\right] \tag{57}$$

$$= \mathbb{E}[\boldsymbol{z}_i^{(a)}]W^Q W^{K^T}(\mathbb{E}[\boldsymbol{z}_k^{(v)}])^T + \mathrm{tr}(\Sigma_{(\boldsymbol{z}_i^{(a)}, \boldsymbol{z}_k^{(v)})}) + \mathbb{E}[\varepsilon_i]W^Q W^{K^T}(\mathbb{E}[\boldsymbol{z}_k^{(v)}])^T \tag{58}$$

The expectation of the attention logit between corrupted tokens is as follows:

$$\mathbb{E}\left[Q(\acute{\boldsymbol{z}}_i^{(a)})K(\acute{\boldsymbol{z}}_j^{(a)})^T\right] \tag{59}$$

$$= \mathbb{E}\left[Q(\boldsymbol{z}_i^{(a)} + \varepsilon_i)K(\boldsymbol{z}_j^{(a)} + \varepsilon_j)^T\right] \tag{60}$$

$$= \mathbb{E}[\boldsymbol{z}_i^{(a)}]W^Q W^{K^T}(\mathbb{E}[\boldsymbol{z}_j^{(a)}])^T + \mathrm{tr}(\Sigma_{(\boldsymbol{z}_i^{(a)}, \boldsymbol{z}_j^{(a)})}) + \mathbb{E}[\boldsymbol{z}_i^{(a)}]W^Q W^{K^T}(\mathbb{E}[\varepsilon_j])^T \tag{61}$$

$$+ \mathbb{E}[\varepsilon_i]W^Q W^{K^T}(\mathbb{E}[\boldsymbol{z}_j^{(a)}])^T + \mathbb{E}[\varepsilon_i]W^Q W^{K^T}(\mathbb{E}[\varepsilon_j])^T \tag{62}$$

We consider the change in attention logits after corruption, e.g., $Q(\acute{z}_i^{(a)})K(z_k^{(v)})^T - Q(z_i^{(a)})K(z_k^{(v)})^T$ and $Q(\acute{z}_i^{(a)})K(\acute{z}_j^{(a)})^T - Q(z_i^{(a)})K(z_j^{(a)})^T$. The expectation of $Q(\acute{z}_i^{(a)})K(z_k^{(v)})^T - Q(z_i^{(a)})K(z_k^{(v)})^T$ is:

$$\mathbb{E}\big[Q(\acute{z}_i^{(a)})K(z_k^{(v)})^T - Q(z_i^{(a)})K(z_k^{(v)})^T\big] = \mathbb{E}\big[Q(z_i^{(a)} + \varepsilon_i)K(z_k^{(v)})^T\big] - \mathbb{E}\big[Q(z_i^{(a)})K(z_k^{(v)})^T\big] \tag{63}$$

$$= \mathbb{E}[\varepsilon_i]W^Q W^{K^T}(\mathbb{E}[z_k^{(v)}])^T \tag{64}$$

The expectation of $Q(\acute{z}_i^{(a)})K(\acute{z}_j^{(a)})^T - Q(z_i^{(a)})K(z_j^{(a)})^T$ is:

$$\mathbb{E}\big[Q(\acute{z}_i^{(a)})K(\acute{z}_j^{(a)})^T - Q(z_i^{(a)})K(z_j^{(a)})^T\big] \tag{65}$$

$$= \mathbb{E}\big[Q(z_i^{(a)} + \varepsilon_i)K(z_j^{(a)} + \varepsilon_j)^T\big] - \mathbb{E}\big[Q(z_i^{(a)})K(z_j^{(a)})^T\big] \tag{66}$$

$$= \mathbb{E}[z_i^{(a)}]W^Q W^{K^T}(\mathbb{E}[\varepsilon_j])^T + \mathbb{E}[\varepsilon_i]W^Q W^{K^T}(\mathbb{E}[z_j^{(a)}])^T + \mathbb{E}[\varepsilon_i]W^Q W^{K^T}(\mathbb{E}[\varepsilon_j])^T \tag{67}$$

The combination of linear transformation $W^Q W^{K^T}$ can be decomposed by singular value decomposition into $U\Sigma V^T$, where $\sigma_{max} > 0$ represents the largest singular value of $\Sigma$ (excluding the trivial case of $W^Q W^{K^T}$ is all-zero matrix). Based on the Cauchy-Schwarz inequality, i.e., $|\langle a, b \rangle| \leq \|a\|_2 \|b\|_2$, and the properties of singular values, i.e., $\|aU\Sigma V^T\|_2 <= \sigma_{max}\|a\|_2$, we can calculate the least upper bound of the absolute values of the two expectations mentioned above.

$$\left|\mathbb{E}\big[Q(\acute{z}_i^{(a)})K(z_k^{(v)})^T - Q(z_i^{(a)})K(z_k^{(v)})^T\big]\right| = \left\|\mathbb{E}[\varepsilon_i]W^Q W^{K^T}(\mathbb{E}[z_k^{(v)}])^T\right\|_2 \tag{68}$$

$$\leq \left\|\mathbb{E}[\varepsilon_i]W^Q W^{K^T}\right\|_2 \cdot \|\mathbb{E}[z_k^{(v)}]\|_2 \tag{69}$$

$$\leq \sigma_{max}\|\mathbb{E}[\varepsilon_i]\|_2 \cdot \|\mathbb{E}[z_k^{(v)}]\|_2 \tag{70}$$

Similarly,

$$\left|\mathbb{E}\big[Q(\acute{z}_i^{(a)})K(\acute{z}_j^{(a)})^T - Q(z_i^{(a)})K(z_j^{(a)})^T\big]\right| \tag{71}$$

$$= \left\|\mathbb{E}[\varepsilon_i]W^Q W^{K^T}(\mathbb{E}[z_j^{(a)}])^T + \mathbb{E}[z_i^{(a)}]W^Q W^{K^T}(\mathbb{E}[\varepsilon_j])^T + \mathbb{E}[\varepsilon_i]W^Q W^{K^T}(\mathbb{E}[\varepsilon_j])^T\right\|_2 \tag{72}$$

$$\leq \sigma_{max}\|\mathbb{E}[\varepsilon_i]\|_2 \cdot \|\mathbb{E}[z_j^{(a)}]\|_2 + \sigma_{max}\|\mathbb{E}[z_i^{(a)}]\|_2 \cdot \|\mathbb{E}[\varepsilon_j]\|_2 + \sigma_{max}\|\mathbb{E}[\varepsilon_i]\|_2 \cdot \|\mathbb{E}[\varepsilon_j]\|_2 \tag{73}$$

**Proposition A.3.** *According to Eq. 70 and Eq. 73, we obtain the relationship of least upper bound between the absolute expected attention logit difference of shited-modality-to-unshifted-modality and unshited-modality-to-unshifted-modality as Eq. 74.*

$$\sup\big(\big|\mathbb{E}\big[\underbrace{\text{AL}(\acute{z}_i^{(a)}, z_k^{(v)}) - \text{AL}(z_i^{(a)}, z_k^{(v)})}_{\text{change of cross-modal AL after shift}}\big]\big|\big) < \sup\big(\big|\mathbb{E}\big[\underbrace{\text{AL}(\acute{z}_i^{(a)}, \acute{z}_j^{(a)}) - \text{AL}(z_i^{(a)}, z_j^{(a)})}_{\text{change of intra-shited-modal AL after shift}}\big]\big|\big) \tag{74}$$

$$\text{when } \mathbb{E}[z_k^{(v)}]\|_2 < \|\mathbb{E}[z_i^{(a)}]\|_2 + \|\mathbb{E}[z_j^{(a)}]\|_2 + \|\mathbb{E}[\varepsilon_j]\|_2$$

*Remark* A.4. The relationship in the above least upper bounds indicates that the attention logit between noisy tokens has larger extreme values. This reaches to the conclusion of damaged multi-modal fusion similar to A.2. We can also have the conclusion from Eq.73 that when the magnitude of noise ($\|\mathbb{E}[\varepsilon]\|_2$) is greater, the difference between the extreme values of corrupted intra-modal attention logit and cross-modal attention logit will also be larger. This explains the experimental phenomenon of the heavier the noise in TTA, the worse the performance (Yang et al., 2024).

## B. Single Modality Experiments

Previous TTA methods such as Tent (Wang et al., 2020), ETA (Niu et al., 2022), SAR (Niu et al., 2023), and READ (Yang et al., 2024) utilize techniques like batch re-normalization, entropy minimization, and sample selection. Blindly applying TTA methods to each modality-specific encoder can lead to redundancy or even harm when no distribution shift is present, resulting in overfitting in unshifted modalities that the original model already generalizes to. This challenge renders previous test-time adaptation methods inadequate, particularly those that apply adaptation without considering the disparities between modalities. As shown in Fig. 6, to observe the performance gap in individual modality-specific modules, we perform experiments on single modality datasets with no shift, and discovey the negative transfer (Rosenstein et al., 2005) phenomenon. This mimics the failure of TTA when they operate on modality-specific encoder for unshifted modality.

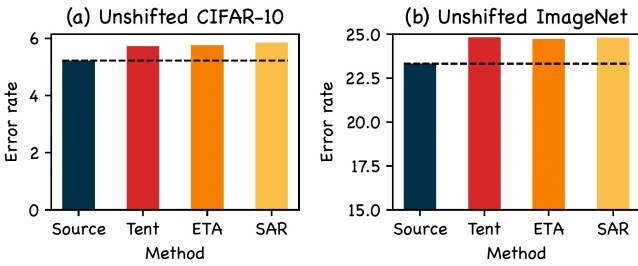

*Figure 6.* The special "negative transfer" phenomenon on the unshifted modality. The introduction of adaptation techniques on the unshifted data results in performance degeration (Source model vs. Tent, ETA, and SAR).

## C. Partial Adaptation Trends

To investigate how the model adapts to synthetic data shifts in specific modalities, we design experiments where we explicitly control shifts in either the video or audio data in Table 6. When synthetic shifts are introduced to the audio data during inference (e.g., Gaussian or traffic noise on audio), the model prioritizes audio adaptation in 62–69% of cases, relying less on video adaptation (31–38%). Conversely, under video shifts (e.g., Gaussian or shot noise on video), the model adapts to video in 53–56% of cases, minimizing reliance on audio adaptation (44–46%). The results are:

*Table 6.* Audio and Video Shift Percentages

| Audio Shift | Percentage | Video Shift | Percentage |
|---|---|---|---|
| Gaussian Noise | 62% | Gaussian Noise | 53% |
| Traffic | 69% | Shot Noise | 56% |

Notably, not all samples with a shifted modality follow the expected route. For example, even with video shifts, 44-47% of predictions still rely on the adapted audio data. We suspect this could be attributed to two reasons: 1) Modality imbalance, where certain modalities exert a greater influence on predictions in multi-modal tasks, makes the model tend to learn from the dominant modalities. 2) Convergence dynamics of the adapter and router. They may not receive sufficient adaptation on one iteration over the test set. It's important to note that making predictions with a partially adapted model is a characteristic of test-time adaptation. This observation highlights a potential area for further research.

## D. Adaptation Details

### D.1. Hyper-parameters

We mainly have the following hyper-parameters: The coefficient and threshold of self-training loss, the softmax temperature, the batch size. We use one set of hyper-parameters for the shift on one modality on each dataset ( we keep the temperature to 0.001 and loss coefficient as 0.5 across all experiments). For Kinetics50-C with video shift, the threshold as 0.9, the batch size as 16. For Kinetics50-C with audio shift, the threshold as 0.9, the batch size as 64. For VGGSound-C with video shift, the threshold as 0.8, the batch size as 128. For VGGSound-C with audio shift, the threshold as 0.8, the batch size as 64. During test-time, our model is updated using the Adam optimizer. We implement the network on a GeForce RTX(TM) 3090 GPU and Intel(R) Core(TM) i9-10900K CPU @ 3.70GHz. For the software information and other experimental settings, please refer to our code https://github.com/chenmc1996/Uni-Modal-Distribution-Shift.

### D.2. Balance Loss

The class balance loss is caculated as:

$$p_{sum} = \sum_i (\hat{p_i}), \tag{75}$$

where $\hat{\boldsymbol{p}_i}$ corresponds to the $i$-th sample. The balance loss, $L_{bal}$, is then defined as its negative entropy:

$$L_{bal} = -\text{entropy}(\boldsymbol{p}_{sum}). \tag{76}$$

This loss serves to regularize the distribution of the model's confidence across different classes, promoting a more even distribution of probabilities.

## E. Sensitivity Test

We avoided heavily tuning the loss coefficient and confidence threshold as it simply follows the self-training loss.

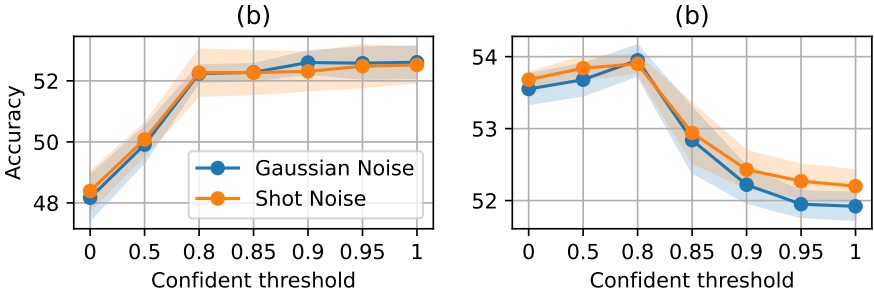

*Figure 7.* Sensitivity test of the on self-training loss on Kinetics50-C and VGGSound-C with video shifts.

The plots in Fig. 7, which vary the confidence threshold for the self-training loss on two dataset. We find that the model can achieve competitive performance with only the balance loss on the Kinetics50-C dataset. However, on the VGGSound-C dataset, removing the self-training loss results in about 2% drops. We also observe consistent performance for varying loss coefficient as shown in Fig. 8.

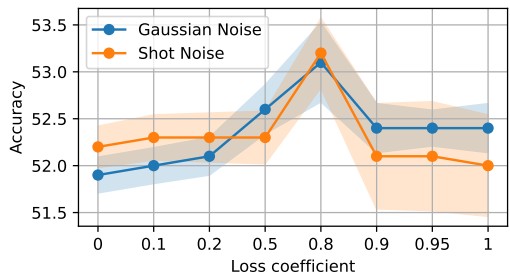

*Figure 8.* Sensitivity test of the on self-training loss on Kinetics50-C with video shifts.

