# OpenReview forum: "Test-Time Selective Adaptation for Uni-Modal Distribution Shift in Multi-Modal Data"
_ICML.cc/2025/Conference — ICML 2025 poster_

### Official Review · Reviewer_5Yv6 · 2025-03-09

**Overall Recommendation:** 1

**Summary:**

This paper address uni-modal distribution shift in multi-modal data, where the distribution shift influences only one modality. They demonstrate that the presence of such shift impedes multi-modal fusion and leads to the negative transfer phenomenon in existing test-time adaptation techniques through theoretical and empirical analyses. Finally, a selective adaptation schema is proposed adapters and a “router” module. Experiments on two datasets highlight its superior performance.



## update after rebuttal

Although the author said they would update the conceptual contribution and adjust claims, the severe misclaim and limited performance improvement in some scenarios make me keep the original score.

**Claims And Evidence:**

The authors clain that "In this research, we define a new practical scenario as uni-modal distribution shift, where the distribution shift influences only one modality, leaving the others unchanged." However, their setting is not new and exactly the same as the previous work [1].

[1] Test-time Adaption against Multi-modal Reliability Bias (ICLR 2024)

**Essential References Not Discussed:**

No

**Experimental Designs Or Analyses:**

The experiments lack autonomous driving tasks such as multimodal segmentation as in [2].

[2] Mm-tta: multi-modal test-time adaptation for 3d semantic segmentation (CVPR 2022)

**Methods And Evaluation Criteria:**

The authors motivate the paper with the application of a self-driving car equipped with complementary camera and LiDAR sensors (Figure 1 and intro). However, they only experiment on action recognition datasets with video and audio. It is important to also validate the method on autonomous driving tasks as in MM-TTA [2].

[2] Mm-tta: multi-modal test-time adaptation for 3d semantic segmentation (CVPR 2022)

**Other Comments Or Suggestions:**

The "router" module is not clearly defined in Section 3. I assume it is equation 7 but the author never mentioned it.

**Other Strengths And Weaknesses:**

Strengths
1. The paper addresses the multimodal test-time adaptation problem, which is a challenging and practical scenario.
2. The paper is well written and easy to follow.
3. The paper provides extensive experiments, showing the effectiveness and versatility of the proposed method.

Major Weaknesses
1. The claimed new practical scenario is not new and exactly the same as the previous work [1].
2. The idea of adapters and router is also not novel and widely used in the literature [3][4].
3. The paper is motivated with the application of a self-driving car equipped with complementary camera and LiDAR sensors (Figure 1 and intro). However, they only experiment on action recognition datasets with video and audio. It is important to also validate the method on autonomous driving tasks as in MM-TTA [2].

[1] Test-time Adaption against Multi-modal Reliability Bias (ICLR 2024)

[2] Mm-tta: multi-modal test-time adaptation for 3d semantic segmentation (CVPR 2022)

[3] Clip-adapter: Better vision-language models with feature adapters (IJCV 2024)

[4] Sparse Mixture-of-Experts are Domain Generalizable Learners (ICLR 2023)

**Questions For Authors:**

How is the method sensitive to alpha in equation 11?

**Relation To Broader Scientific Literature:**

The setting is not new and already discussed as in the previous work [1]. The idea of adapters and router is also not new and widely used in the literature [3][4].

[1] Test-time Adaption against Multi-modal Reliability Bias (ICLR 2024)

[3] Clip-adapter: Better vision-language models with feature adapters (IJCV 2024)

[4] Sparse Mixture-of-Experts are Domain Generalizable Learners (ICLR 2023)

**Theoretical Claims:**

Yes, Proposition 3.2. No issues found.

---

> ### Author Rebuttal · Authors · 2025-04-01
>
> # Response to Reviewer 5Yv6
>
> Thanks for the constructive suggestions! We have addressed each point with careful consideration and revised our work accordingly. Below is our detailed response:
>
> ## W1 [Claims of setting]
> We agree that [1] has explored multi-modal shifts more broadly. However, our work explicitly identifies __uni-modal shifts as a distinct and critical subclass of multi-modal shifts__, which necessitates tailored solutions. Our conceptual contributions lie in:
> * [1] studies shifts in any subset of modalities (e.g., partial or all), formulating to the form of $p_s(x)≠p_t(x)$ as shown in its "Sec. Problem Formulation". Whereas our work specifically models shifts occur in one modality $p_t(x^{(k)}) \neq p_s(x^{(k)})$ and $p_t(x^{(i)}) = p_s(x^{(i)}), \forall i \neq k$. We also discussed why [1] is inefficient in our setting in the related work.
> * We further highlight ambiguities of terminology in existing literature, e.g., ref [1] and [2] both focus on "multi-modal shift". However, [2], different from [1], studies "shifts happens in all modalities".  To resolve this inconsistency, we introduce a precise definition of uni-modal shifts to differentiate them from broader multi-modal shifts.
> * Most importantly, we put our emphasis on "__uni__" is we believe its broad practical implication, as we stated in the introduction, many real-world corrupting factors impact only specific modalities.
>
> We have reframed our conceptual contribution as __identifying the unique challenges of uni-modal shift via theoretical and empirical analysis__, while emphasizing its practical significance.
>
> ## W2 [Novelty of methods]
> Building on W1, our analysis reveals that uni-modal shifts undermine cross-modal fusion and induce negative transfer.
> While adapters and routing mechanisms are established concepts, our innovation lies in their _specific integration to tackle the unique challenges of uni-modal distribution shift_:
> ### Residual Adapter Design
> Unlike standard adapters, our architecture explicitly decouples shift-agnostic base features from shift-specific components through residual connections as in Eq. 8, enabling targeted adaptation without catastrophic forgetting.
> ### Shift-Aware Router
> The shift-aware router introduces a gating mechanism that dynamically selects adapters based on modality-specific shift detection, a crucial capability missing in prior routing approaches.
>
> ## W3 [Experiments for autonomous driving tasks]
> With due respect, we offer the following clarifications:
> While we use autonomous driving as an illustrative example, our experiments span diverse tasks (action recognition on Kinetics50, event recognition on VGGSound) with applications extending beyond autonomous driving.
>
> We acknowledge the need for diverse multi-modal shift benchmarks (though most current multi-modal TTA research focuses on the two datasets, Kinetics50 and VGGSound, with various multi-modal shifts). However, reproducibility challenges with MM-TTA (unreleased code) prompted us to validate our approach on other datasets. We choose CMU-MOSI→CMU-MOSEI, real-world sentiment analysis datasets with possible modality-specific shifts: Both datasets include audio, text, and video data. The CMU-MOSEI paper highlights factors such as varying face detection methods that can contribute to visual distribution shifts. For instance, some clips may be more face-centered, indicating a video shift while other modalities remain largely unchanged. To support this claim, we calculate the Maximum Mean Discrepancy (MMD) between each modality to measure their discrepancies:
>
> |Modality|MMD|
> |-|-|
> |audio|0.0042|
> |vision|0.1379|
> |text|0.0105|
>
> The significant vision shift (14× larger than audio/text) mirrors real-world scenarios where single modalities shift. Our experiments on this dataset aim to: 1) provide evidence that many real-world shifts resemble uni-modal shifts and 2) further validate the effectiveness of our method. We then simply apply our "selective adaptation" architecture on a recent work CASP (due to the difference between tasks, we choose to swiftly adopt our core idea to the existing codebase to present the performance in a short rebuttal period). The following are the results:
>
> |Model|ACC ↑|F1 ↑|MAE ↓|
> |-|-|-|-|
> |GC (CVPR 22)|67.12|67.40|1.22|
> |RF (ICLR 24)|66.84|67.39|1.27|
> |CASP (AAAI 25)*|67.02|67.05|1.34|
> |Ours|__67.74__|__67.73__|__0.85__|
> *reproduced using its official code
>
> These results help demonstrate our method's effectiveness on more diverse tasks.
>
> ## S1 [term router]
> We apologize for the oversight. We have strengthened its description in Sec. 3.5 Selective Adaptation, where we detail its role in computing gating weights based on shift severity.
>
> ## Q1
> Please check our reply to Reviewer UDgQ W3 [Sensitivity of hyperparameter].
>
> We thank the reviewer for the constructive feedbacks, which has strengthened our paper. We remain open to further revisions.

---

> > ### Comment · Reviewer_5Yv6 · 2025-04-07
> >
> > I want to thank the authors for the rebuttal and most of my concerns are addressed. However, I still have concerns on W1 regarding the difference between your setup and the setup in READ [1]. Although the name of [1] is "Multimodal reliability bias", if I understand correctly, it addresses the exact unimodal bias as in your paper (only uni-modal shifts).  For example, in Figure 2 in [1], they have corrupted video and clean audio. For all experiments in [1], they also only focus on unimodal bias (corrupted video in Table 1 and 3, and corrupted audio in Table 2). The Table 1-3 in your paper exactly follow that in [2] and I can't see any differences. Therefore, the claimed new practical scenario is not convincing.
> >
> > Besides, the method proposed in this paper doesn't show significant superiority compared to [1]. For example, in Table 2, it achieves the same average accuracy as READ. In Table 3, it only surpasses READ 0.4 on Kinetics50-C.

---

> > > ### Author Response · Authors · 2025-04-08
> > >
> > > # Second Response to Reviewer 5Yv6
> > >
> > > ## Revisions to conceptual contribution
> > > Thank you for your constructive feedback and for highlighting the need to clarify our contributions relative to READ [1]. We have carefully revised the manuscript to address potential confusions and better emphasize our core insights.
> > >
> > > **Key Revisions:**
> > > 1. **Updated Conceptual Contribution:**
> > >    We revised our claim from *“identifying a novel scenario”* to:
> > >    > *We identify the unique challenges of uni-modal shift in multi-modal data through theoretical analysis (i.e., large fluctuations in cross-modal attention) and empirical analysis (i.e., negative transfer).*
> > >
> > >    This shift underscores that our focus is not solely on defining the scenario (which READ [1] preliminarily explores) but on rigorously characterizing its **underlying challenges**, which READ does not address.
> > >
> > > 2. **Corresponding Adjustments of Claims:**
> > >    - **Abstract:** Changed *“define a new practical scenario”* to *“explore the under-explored practical scenario”* to position our work as building on existing studies while advancing new insights.
> > >    - **Introduction:**
> > >      - Revised *“For the first time, we term…”* to *“we term this overlooked shift…”* to acknowledge READ’s prior experimentation.
> > >      - Added a clarification:
> > >        > *“While READ [1] investigates multi-modal shifts and includes preliminary experiments on uni-modal shifts, it does not address their unique challenges (e.g., instability in cross-modal attention or negative transfer during adaptation). For instance, READ reduces attention to corrupted modalities but does not adaptively repair or utilize them, leaving these issues unresolved.”*
> > >
> > > **Conclusion:**
> > > Our revisions aim to explicitly differentiate our theoretical/empirical contributions on uni-modal shift (e.g., analyzing attention instability and negative transfer) from READ’s broader scope. We appreciate your thoughtful critique, which strengthened our framing. Once again, we earnestly request that the reviewer re-assess our contribution with these updates in mind.
> > >
> > > ## Performance improvement
> > >
> > > Thank you for raising this important point. We appreciate the opportunity to clarify the nuances behind the experimental results and highlight the broader significance of our method.
> > >
> > > **Addressing Modality Imbalance in Results:**
> > > The performance differences in Tables 2 and 3 stem from the **inherent modality imbalance** in the datasets:
> > > - **Kinetics50-C** is video-dominant (as reflected in its video-centric data design), so shifts in the *non-dominant* audio modality leave limited room for improvement.
> > > - **VGGSound-C** is audio-dominant (evident from its audio-focused curation), so shifts in the *non-dominant* video modality similarly constrain gains.
> > >
> > > Despite these dataset biases, our method achieves a **1.17% average improvement** over READ [1] across all benchmarks. More critically, our framework **selectively adapts only the shifted modality**, ensuring no degradation to the unshifted (dominant) modality.
> > >
> > > **Broader Implications:**
> > > These results underscore a key insight: real-world multimodal systems often exhibit **imbalanced modality reliance**, and shifts on "weaker" modalities demand delicate adaptation strategies. READ [1], while effective in reweighting modalities, does not explicitly address this challenge. Our method’s *selective adaptation* ensures robustness to shifts on *any* modality (dominant or non-dominant) without cross-modal interference, making it more versatile for practical deployment.
> > >
> > > In conlusion, we agree that improvements on non-dominant shifts (e.g., audio in Kinetics50-C) may appear modest, but this reflects the inherent limitations of imbalanced datasets rather than methodological shortcomings. Our framework’s ability to adapt *safely and selectively*—even in constrained scenarios—highlights its value for real-world applications where modality shifts are unpredictable and imbalanced. Thank you for your insightful critique, which allowed us to better contextualize these results.

---

### Official Review · Reviewer_UDgQ · 2025-03-13

**Overall Recommendation:** 3

**Summary:**

This paper addresses the challenge of uni-modal distribution shifts in multi-modal learning, where only one modality experiences distribution changes at test time. The authors propose a selective adaptation framework comprising modality-specific lightweight adapters and a learnable router to dynamically activate adaptation for the shifted modality. Theoretical analysis examines how uni-modal shifts disrupt cross-modal attention mechanisms, arguing that intra-modal attention logit variance increases more significantly than cross-modal variance under additive noise assumptions. Experiments on Kinetics50 and VGGSound datasets with synthetic corruptions (e.g., noise, blur) demonstrate improved accuracy over existing test-time adaptation methods.

**Claims And Evidence:**

The primary claims of the paper are that:
1.Uni-modal distribution shifts degrade multi-modal fusion by amplifying intra-modal attention variance.
2.Existing TTA methods suffer from negative transfer when applied to unshifted modalities.
3.The proposed selective adaptation using modality-specific adapters and routers improves robustness to such shifts.

The authors conducted experiments on the Kinetics50 and VGGSound datasets, applying various uni-modal offsets such as noise and blur. The results show that existing TTA methods struggle with uni-modal offsets, but the improvements of the proposed selective adaptation methods are not always significant. Some baseline methods remain competitive in some cases, and the effectiveness of the method varies depending on the dataset and type of offset. The theoretical analysis outlines the impact of uni-modal offsets on multimodal fusion, but the practical implications remain somewhat abstract and lack a deep connection to real-world applications.

**Essential References Not Discussed:**

To the best of my knowledge, the authors have cited several key works in TTA and multi-modal learning

**Experimental Designs Or Analyses:**

The experiments are conducted on two multi-modal datasets (Kinetics50 and VGGSound) with uni-modal corruptions (e.g., noise, blur). However, the evaluation is limited as it does not extend to other types of multi-modal datasets or compare on multiple multi-modal pre-trained models. Additionally, it lacks a sensitivity analysis of the key hyperparameter α (Eq. 11), significantly restricting insights into the method’s broader applicability.

**Methods And Evaluation Criteria:**

The paper proposes a selective adaptation framework that incorporates modality-specific lightweight adapters and a learnable router to determine which modality to adapt during test time. The router employs a Gumbel-softmax mechanism and the model is updated via self-training with pseudo-labels. Evaluation is conducted on standard multi-modal benchmarks (Kinetics50 and VGGSound) under uni-modal shifts such as noise and blur, with classification accuracy as the main metric. Although Experiment Q3 provides some analysis on computational efficiency by reporting parameter counts and runtime, the evaluation remains basic and does not thoroughly address scalability or potential trade-offs in more challenging scenarios.

**Other Comments Or Suggestions:**

I suggest exploring more diverse and complex datasets to test the method's scalability. Additionally, discussing potential limitations or failure cases of the method would provide a more balanced view.
The repeated use of the variable τ (Eq. 7 and 10) is confusing.

**Other Strengths And Weaknesses:**

Strengths:
1.This paper presents an interesting real-world problem, uni-modal shifts, which is relevant for multi-modal systems deployed in dynamic environments.
2.The proposed method effectively combines modality-specific adapters and a learnable router, offering a promising solution to the problem in the context of audio and video modalities.
Weaknesses:
1.The novelty of the approach is somewhat limited, as adapters and routing are well-established concepts in the literature.
2.The experiments are relatively limited, focusing mainly on the video and audio modalities using two datasets. This does not adequately demonstrate the method's applicability to a broader range of modality combinations or more diverse multi-modal datasets.
3.The sensitivity analysis of the key hyperparameter α (Eq. 11) is notably missing, which limits understanding of how changes in this parameter affect the model's performance under different conditions.

**Questions For Authors:**

1.How does the router avoid mode collapse (e.g., always selecting one modality)?
2.The method seems to be strictly designed for handling single-modal shifts; how would it address cases where multiple modalities experience shifts?
3.Could you provide the performance of the method on multi-modal datasets with different modality combinations?

**Relation To Broader Scientific Literature:**

The paper positions its contribution within the broader context of test-time adaptation and multi-modal learning. It correctly highlights the limitations of existing methods in handling uni-modal adaptation for audio and video, but it would be more helpful if more comparisons or discussions were provided on models handling other modality data.

**Theoretical Claims:**

This paper provides a sufficient theoretical analysis of how uni-modal transitions affect multimodal fusion, particularly self-attention mechanisms. However, the theoretical insights can be further extended to explain the generalizability of the approach to other types of transitions or scenarios beyond experimental testing.

---

> ### Author Rebuttal · Authors · 2025-04-01
>
> # Response to Reviewer UDgQ
>
> Thanks for the reviewer’s positive comments and constructive suggestions! We have carefully considered each point and revised our work accordingly. Below is our detailed response:
>
> ## W1 [Novelty of method]
>
> We emphasize that our design offers a straightforward solution to the novel challenge of uni-modal shift. While adapters and routing mechanisms are established concepts, our innovation lies in _their specific integration to tackle uni-modal distribution shifts—a critical challenge we analyze theoretically and validate empirically_. Our methodological contributions include:
> ### Residual Adapter Design
> Unlike standard adapters, our architecture explicitly decouples shift-agnostic base features from shift-specific components through residual connections as in Eq. 8, enabling targeted adaptation without catastrophic forgetting.
> ### Shift-Aware Router
> The shift-aware router introduces a gating mechanism that dynamically selects adapters based on modality-specific shift detection, a crucial capability missing in prior routing approaches.
>
> This integration uniquely addresses scenarios where only one modality shifts, a common real-world issue under-explored in prior work.
>
> ## W2 & Q3 [Experiments on multi-modal datasets with different modality combinations]:
>
> First, we would like to bring to the reviewer’s attention that most current multi-modal TTA research focuses on Kinetics50 and VGGSound and with various multi-modal shifts. Nevertheless, we acknowledge the importance of more diverse multi-modal distribution shift benchmarks.
> Therefore, we further investigate a real-world distribution shift within multi-modal datasets by examining the transfer from CMU-MOSI to CMU-MOSEI, which includes audio, text, and video data for sentiment analysis. The CMU-MOSEI paper highlights factors such as varying face detection methods that can contribute to visual distribution shifts. Our experiments on this dataset aim to: 1). provide evidence that many real-world shifts resemble uni-modal shifts and 2). further validate the effectiveness of our method. For instance, some video clips may be more face-centered, indicating a video shift while other modalities remain largely unchanged. To support this claim, we calculate the Maximum Mean Discrepancy distance between each modality to measure their discrepancies:
>
> |Modality|MMD|
> |-|-|
> |audio|0.0042|
> |vision|0.1379|
> |text|0.0105|
>
> The significant vision shift (14× larger than audio/text) mirrors real-world scenarios where single modalities shift.
> We then simply apply our "selective-adaptation" architecture on a recent work CASP (due to the difference between tasks, we choose to swiftly adopt our core idea to the existing codebase to present the performance in a short rebuttal period). The following is our results:
>
> |Model|ACC ↑|F1 ↑|MAE ↓|
> |-|-|-|-|
> |GC (CVPR 22)|67.12|67.40|1.22|
> |RF (ICLR 24)|66.84|67.39|1.27|
> |CASP (AAAI 25)*|67.02|67.05| 1.34|
> |Ours|__67.74__|__67.73__|__0.85__|
> *reproduced using its official code
>
> These results help demonstrate our method's effectiveness on broader ranges of modality combinations or more diverse multi-modal datasets.
>
> ## W3 [Sensitivity of hyperparameter]
> We thank the reviewer for this observation. We avoided heavily tuning the loss coefficient as it simply follows the self-training loss. Nevertheless, we do agree that a sensitivity analysis of α would provide technical insights. We will include this analysis in the revision to strengthen the empirical validation of our method. The results are as follows:
>
> |α|0.0|0.1|0.2|0.5|0.8|0.9|0.95|1.0|
> |-|-|-|-|-|-|-|-|-|
> |Accuracy|51.9|52.0|52.1|52.6|53.1|52.4|52.4|52.4|
>
> In conclusion, we observe robust performance for α∈[0,1]:
>
>
> ## Q1 [Model collapse]
> + Probabilistic sampling (Eq. 6) in Gumbel-Softmax introduces randomness to prevent deterministic routing.
> + In each of our experiments, only one modality is shifted at a time, so the 'model collapse', i.e., consistently selecting one modality is not necessarily detrimental.
>
> ## Q2 [Shifts of multiple modalities]
> Good question! Although our setting aligns with many practical applications, real-world environments often involve more complex combinations of distribution shifts across modalities. This poses significant challenges and requires further research. For now:
> + The selection of adapters is conducted in a soft way (Eq. 8), i.e., they may function simultaneously and allow partial contributions from multiple modalities.
> + We have added a limitation section at the last of our manuscript with the discussion regarding our setting’s limitation of encountering more complex shifts within multiple modalities or continual shift environments. Please check our reply to Reviewer dkfb.
>
> ## Suggestions [misuse of symbol]
> We have corrected the issue. We thank your meticulous review!
>
> We thanks the Reviewer's suggestions that help strengthen our paper’s demonstration of its applicability.

---

### Official Review · Reviewer_dkfb · 2025-03-13

**Overall Recommendation:** 3

**Summary:**

This paper proposes a novel approach to handling multi-modal test-time adaptation when only one modality undergoes distribution shift. The authors introduce the concept of uni-modal distribution shift, highlighting its adverse effects on multi-modal fusion and the potential for negative transfer. To address this issue, the paper proposes a selective adaptation framework that integrates modality-specific adapters with a routing mechanism that dynamically determines which modality requires adaptation. The effectiveness of the proposed method is validated through extensive empirical evaluations on datasets exhibiting uni-modal shifts, demonstrating superior performance compared to existing TTA methods.

**Claims And Evidence:**

1. Uni-modal distribution shift disrupts multi-modal fusion. Supported by both theoretical analysis and empirical results, demonstrating performance degradation when conventional TTA methods are applied indiscriminately across modalities.
2. Selective adaptation enhances test-time robustness. Experimental findings indicate that the proposed router-based approach mitigates negative transfer and improves adaptation efficacy.
3. The proposed method surpasses state-of-the-art (SOTA) TTA techniques. Performance evaluations on Kinetics50-C and VGGSound-C show consistent gains over Tent, ETA, SAR, MM-TTA, and READ.
4. Selective adaptation is computationally efficient. The paper presents comparative analyses of computational cost, showing that the approach maintains efficiency while improving accuracy.

**Essential References Not Discussed:**

The literature review is comprehensive, but the discussion could be expanded to include, (1) Adaptive batch normalization techniques (AdaBN, BN-statistics adaptation) as alternative strategies for domain adaptation. (2) Continual learning paradigms that manage selective forgetting, which may offer insights into selective adaptation mechanisms.

**Experimental Designs Or Analyses:**

The experimental methodology is well-structured, including (1) Comparative evaluations across multiple corruption types (e.g., Gaussian noise, motion blur, weather conditions). (2) Ablation studies to assess the contributions of key components, such as the router, Gumbel-softmax selection, and test-time inference schema. And (3) computational efficiency assessments, demonstrating the balance between accuracy improvements and resource consumption.

**Methods And Evaluation Criteria:**

The study utilizes the Kinetics50-C and VGGSound-C datasets, incorporating 21 types of uni-modal distribution shifts. Evaluation metrics include classification accuracy across different corruption types and computational overhead analysis. The experimental setup is well-aligned with real-world scenarios, as it effectively models cases where only one modality undergoes degradation.

**Other Comments Or Suggestions:**

N/A

**Other Strengths And Weaknesses:**

**Strengths:**

1. Introduces uni-modal distribution shift, a novel and practically relevant problem.

2. The router-adapter mechanism effectively mitigates negative transfer.

3. Strong empirical validation demonstrating state-of-the-art performance.

4. The paper is well-written and easy to follow.

**Weaknesses:**

1. The theoretical justification of the router’s effectiveness could be further developed.

2. Long-term adaptation stability remains unexplored. [A]

[A] A Probabilistic Framework for Lifelong Test-Time Adaptation.

**Questions For Authors:**

See weakness above.

**Relation To Broader Scientific Literature:**

This work builds upon prior advancements in TTA (Tent, ETA, SAR) by introducing a modality-selective adaptation approach. It also extends research in multi-modal fusion by addressing negative transfer through targeted adaptation. Compared to MM-TTA and READ, which apply adaptation broadly across modalities, this work is among the first to tackle selective adaptation for uni-modal shifts.

**Theoretical Claims:**

This paper provides a theoretical examination of the impact of uni-modal distribution shifts on self-attention-based multi-modal fusion. The derivations are mathematically sound, illustrating how attention weight distributions are disrupted under such shifts.

---

> ### Author Rebuttal · Authors · 2025-04-01
>
> # Response to Reviewer dkfb
>
> Thanks for the reviewer’s positive comments and constructive suggestions! We have carefully considered each point and revised our work accordingly. Below is our detailed response:
>
> ## W1 [Theoretical justification router’s effectiveness]:
> We agree that the theoretical underpinning of our router mechanism could be further developed. Our current understanding is in line with modality-specific attention mechanisms: The router (followed by cross-attention fusion) essentially learns to enhance attention computation conditioned on shift detection statistics (Eq. 8). Besides, we notice that Li et al. [1] give a theoretical study on this direction very recently, showing how the MoE-like structure help in multi-tasks or multi-domain scenario.  Drawing on the ideas presented in the article, we anticipate that theories related to 'shift-specific MoE' could serve as a starting point.
>
> While time constraints prevent full theoretical development in this submission, we would welcome the reviewer’s suggestions on specific theoretical directions to prioritize in our revision or future work.
>
> [1] Hongbo Li, et al. "Theory on Mixture-of-Experts in Continual Learning." The Thirteenth ICLR. 2025
>
> ## Suggestion regarding literature review & W2 [Long-term adaptation]:
> Thanks for bringing the related works to our attention.
> In terms of the Adabn, the reviewer might overlook our discussion in the related work where we discussed Adabn:
> ```
> The Adabn (Li et al., 2017) recomputes the batch statistics for every test batch ....
> ```
>
> As for the long-term adaptation stability, while our primary contribution remains in multi-modal shift rather than continual adaptation, we fully agree this is critical for real-world deployment. We now explicitly position our work as complementary to lifelong adaptation research: We have added a limitation section at the last of our manuscript as follows with the discussion regarding the lifelong/continual shift over time mentioned in the ref [A] along with our setting’s limitation of encounter more complex shifts within multiple modalities:
>
> ```
> Section 6. Limitations
> While our work provides a novel view for the unique uni-modal distribution shifts in multi-modal test-time adaptation, it is crucial to acknowledge several limitations.
>
> First, our current formulation primarily focuses on scenarios where a distribution shift occurs in only one modality at test time. Although this setting aligns with many practical applications, real-world environments often involve more complex combinations of distribution shifts across modalities. For instance, cases where two modalities experience concurrent shifts while the third remains unchanged, or shifts with varying magnitudes, pose significant challenges. Our theoretical and empirical analyses reveal that existing methods, including our proposed approach, struggle to disentangle conflicting signals from multiple shifted modalities, leading to suboptimal fusion and adaptation performance. This limitation underscores the need for more sophisticated mechanisms to diagnose and disentangle multi-modal shifts dynamically.
>
> Besides, our method operates under the assumption of single mini-batch test-time adaptation, where adaptation occurs incrementally on small, temporally coherent batches. While this setup is practical for scenarios with transient shifts, it does not account for long-term or continual distribution shifts that evolve over extended periods [1-2]. For example, in real-world deployments, modalities may drift gradually or exhibit recurring shifts, requiring adaptation strategies that balance plasticity (to learn new patterns) with stability (to retain prior knowledge). Our method does not explicitly address catastrophic forgetting or the accumulation of adaptation errors over time, which remain critical open challenges.
>
> These limitations, however, highlight promising directions for future research. The failure of existing methods in multi-shift scenarios motivates the development of adaptive routing mechanisms that can scale to higher-order modality combinations and dynamically prioritize shifts based on their severity. Similarly, extending our selective adaptation idea to continual or long-term settings could involve memory-augmented architectures or meta-adaptation strategies to stabilize performance over time. We hope our work inspires the community to explore these frontiers, ultimately advancing the robustness of multi-modal systems in increasingly complex real-world environments.
>
> [1] Wang, Qin, et al. "Continual test-time domain adaptation." Proceedings of the IEEE/CVF Conference on CVPR. 2022.
> [2] Brahma, Dhanajit, and Piyush Rai. "A probabilistic framework for lifelong test-time adaptation." Proceedings of the IEEE/CVF Conference on CVPR. 2023.
> ```
>
> We appreciate the reviewer's suggestions, which have strengthened our paper's literature review and clarified its limitations.

---

### Official Review · Reviewer_hbmC · 2025-03-14

**Overall Recommendation:** 3

**Summary:**

The paper tackles the problem of multi-modal adaptation—the existing method of test time Domain adaptation struggles to adapt to other modalities. The authors propose to highlight this phenomenon and tackle this challenge with a "router" enabling the selection of the adaptation if needed. The selective adaptation method is tested over two datasets and compared to 4 other SOTA methods.

**Claims And Evidence:**

Plots and good experiments support the claims of the paper.

**Essential References Not Discussed:**

Seems good.

**Experimental Designs Or Analyses:**

The experiments are appropriately designed and show good performance and efficiency in the proposed method.

**Methods And Evaluation Criteria:**

The method uses some existing methods (i.e., losses) but adds the selective adaptation that makes sense for the problem of heterogeneous domain adaptation. The method is tested over two datasets of multi-modalities.

**Other Comments Or Suggestions:**

In my opinion, the term "router" is misleading and is only used at the beginning of the paper. Maybe using a more transparent term like "Selective Adaptation" or something like that can be better.

**Other Strengths And Weaknesses:**

Strengths:
- The paper is easy to follow.
- The proposition of having a selective adaptation is an excellent idea.
- The experiments are well done and seem trustworthy.

Weaknesses:
- Figure 3 is complex to read at the beginning of the paper. It is not clear what the role of the router at the beginning. It can be nice to have a more detailed figure caption and marble put a color legend or something like that.
- Study of router's adaptation weights (cf questions)

**Questions For Authors:**

- You do not study the selective adaptation in your experiments. In the ablation study, you show that the selection is crucial. I would like to know whether the proportion of data is getting adapted or not. Maybe with the visualization of the data, we can understand which type of data needs to be adapted or not.

**Relation To Broader Scientific Literature:**

The paper is well related to broader literature.

**Theoretical Claims:**

I didn't have time to check the proof thoroughly.

---

> ### Author Rebuttal · Authors · 2025-04-01
>
> # Response to All Reviewers, AC and SAC
> Thanks for all Reviewers' valuable suggestions and the efforts of AC and SAC.
>
> Reviewer dkfb, UDgQ and 5Yv6 all acknowledge the studied problem (we quote: "_novel and practically relevant_", "_interesting real-world problem_" and "_challenging and practical scenario_").
> Reviewer hbmC, dkfb, and UDgQ all give positive comments on our methodology (we quote: "_excellent idea_", "_effectively mitigates negative transfer_" and "_promising solution_").
> Furthermore, our theoretical and empirical results are appreciated by all Reviewers (we quote: "_sufficient theoretical analysis_", "_mathematically sound_", "_trustworthy_", “_strong empirical validation_” and “_extensive experiments_”).
> Lastly, all Reviewers find our paper well presented.
> We'll integrate the Reviewers' suggestions into our revision. Next, we respond to all of the questions.
>
> # Response to Reviewer hbmC
>
> ## W1 [Clarity of Figure 3]
> We appreciate this constructive feedback. To enhance clarity, we have expanded the figure caption to explicitly describe the router’s role and the architecture’s flow. The new caption reads:
> ```
> Fig. 3: The architecture of our model. The original multi-modal data is processed through modality-specific encoders to extract feature representations. A router then dynamically decides whether to adapt each modality. For instance, in the case of video corruption, the router prioritizes video adaptation (red path) while preserving audio features (blue path). Finally, the extracted features are fused and passed to the prediction head.
> ```
>
> ## W2 & Q1 [Study of router's adaptation weights]
> We thank the reviewer for raising this insightful question. Indeed, selective adaptation is a critical aspect of our framework, and we have conducted additional analyses to address this point. To investigate how the model adapts to synthetic data shifts in specific modalities, we design experiments where we explicitly control shifts in either the video or audio data. Our key findings are as follows:
> ### Modality-Specific Adaptation Routes:
> When synthetic shifts are introduced to the audio data during inference (e.g., Gaussian or traffic noise on audio), the model prioritizes audio adaptation in  62–69% of cases, relying less on video adaptation (31–38%). Conversely, under video shifts (e.g., Gaussian or shot noise on video), the model adapts to video in 53–56% of cases, minimizing reliance on audio adaptation (44–46%). The results are:
> | Audio shift | Percentage |
> |-|-|
> | Gaussian Noise  | 62%  |
> | Traffic | 69%  |
>
> | Video shift | Percentage |
> |-|-|
> | Gaussian Noise  | 53%  |
> | Shot Noise  | 56%  |
>
> ### Partial Adaptation Trends:
>
> Notably, not all samples with a shifted modality follow the expected route. For example, even with video shifts, 44-47% of predictions still rely on the adapted audio data. We suspect this could be attributed to two reasons: 1) Modality imbalance, where certain modalities exert a greater influence on predictions in multi-modal tasks, makes the model tend to learn from the dominant modalities. 2) Convergence dynamics of the adapter and router. They may not receive sufficient adaptation on one iteration over the test set. It's important to note that making predictions with a partially adapted model is a characteristic of test-time adaptation. This observation highlights a potential area for further research.
>
> In conclusion, these results highlight the model’s ability to dynamically prioritize modalities based on their reliability under distribution shifts. We will include these findings and detailed visualizations in the revised manuscript, along with a discussion of potential explanations.
>
> ### Suggestion [Term "router"]
> We acknowledge that the "router" could cause confusion and the adaptation selector could be more straightforward. We’ll make the change accordingly in the revision.
>
> Thank the reviewer again for prompting the analysis. We believe it strengthens the paper’s contribution by demonstrating the adaptability and interpretability of our framework.

---

> > ### Comment · Reviewer_hbmC · 2025-04-04
> >
> > I thank the authors for their answer and additional experiments. As said by reviewer 5Yv6, your proposition is very close to the READ method. I rechecked the results, and I wonder if the improvement in the score is significant. Can you provide a statistical test to compare your method and READ?

---

> > > ### Author Response · Authors · 2025-04-06
> > >
> > > # Second Response to Reviewer hbmC
> > > Thanks for the reply and the Reviewer's interest in our work!
> > >
> > > We appreciate the opportunity to clarify the distinction between our work and the READ method [1]. Our contributions are fundamentally distinct in both conceptual framing and practical implications. Below, we detail the key distinctions to highlight the uniqueness of our contributions:
> > >
> > > + READ [1] formulates shifts as occurring in any subset of modalities (including partial or full combinations), framed broadly as $p_s(x)≠p_t(x)$. In contrast, our work isolates and rigorously defines uni-modal shifts as shifts occurring exclusively in a single modality $p_t(x^{(k)}) \neq p_s(x^{(k)})$ and $p_t(x^{(i)}) = p_s(x^{(i)}), \forall i \neq k$. This distinction is not merely semantic: as discussed in Sec. 2.2 and our experiments, READ’s general formulation leads to inefficiency in our setting. Since READ aims to reduce attention on shifted modalities, it fails to fully utilize the valuable information present in those modalities.
> > >
> > > + Existing literature conflates terms "multi-modal shift" (e.g., [1] vs. [2]). However, [2] assumes shifts occur in all modalities, while [1] allows shifts in any subset. To resolve this inconsistency, we introduce uni-modal shift as a precise, standalone category (Sec. 1.1).
> > > By doing so, we contribute to clarifying the taxonomy of distribution shifts, which has been muddled in previous research.
> > >
> > > + As emphasized in our introduction, we believe uni-modal shifts are pervasive in real-world scenarios (e.g., many corrupting factors only cause modality-specific shfits).  This will enable more targeted research and the development of more effective solutions for real-world problems where such shifts occur.
> > >
> > > To further clarify our proposition, we have revised the contribution statement, focusing on __identifying the unique challenges of uni-modal shift via theoretical and empirical analysis__.
> > >
> > > [1] Test-time Adaption against Multi-modal Reliability Bias (ICLR 2024)
> > >
> > > [2] Mm-tta: Multi-modal Test-time Adaptation for 3d Semantic Segmentation (CVPR 2022)
> > >
> > > In response to the request for a statistical test to compare our method with READ, we are pleased to provide the following results. To demonstrate the performance differences, we conducted 5 runs for both our method and READ and calculated the standard deviation (std) for each run. Additionally, we performed McNemar's test and obtained the corresponding p-values. The detailed results are presented in the table below:
> > >
> > > Comparisons with READ on Kinetics50-C benchmark with corrupted video data.
> > > | |Gauss.| Shot | Impul.|
> > > |-|-|-|-|
> > > |READ* | 49.13 ± 0.3 | 49.54 ± 0.25 | 49.15 ± 0.31|
> > > |Ours | 52.6 ± 0.37 |52.31 ± 0.62 |51.96 ± 0.85|
> > > |McNemar's test| 9.3e-07 | 0.0007 | 0.0006|
> > > *reproduced using its official code
> > >
> > > Comparisons with READ on VGGSound-C benchmark with corrupted audio data.
> > > | |Gauss.| Traff. | Crowd|
> > > |-|-|-|-|
> > > |READ* |  39.99 ± 0.39 | 28.93 ± 0.34 | 26.67 ± 0.31|
> > > |Ours | 41.5 ± 0.14 |31.8 ± 0.29 |30.93 ± 0.30|
> > > |McNemar's test| 9.1e-08 | 3.5e-27 | 2.6e-53 |
> > > *reproduced using its official code
> > >
> > > The low p-values obtained from McNemar's test (<0.5) indicate that the differences in performance between our method and READ are statistically significant. The standard deviations show the variability across the 5 runs, highlighting the stability of both methods. Overall, these results confirm the superiority of our approach and validate the statistical significance of the performance gains.
> > >
> > > We sincerely appreciate the reviewer's feedback and hope that this response clearly elucidates the novel concepts of our work and the statistical significance of the performance improvements we have achieved.

---

### Decision · Program_Chairs · 2025-05-01

**Decision:**

Accept (poster)

**Comment:**

This paper addresses unimodal distribution shifts in multimodal data, proposing a selective adaptation framework with modality-specific adapters and a router module. Reviewers acknowledge its practical relevance, noting strong empirical performance on Kinetics50-C and VGGSound-C datasets, where it outperforms baselines like Tent and READ in scenarios with single-modality corruption.

Theoretical analysis highlights how unimodal shifts disrupt cross-modal attention, and the router-adapter design effectively mitigates negative transfer. However, concerns persist: limited novelty is raised, as the scenario and adapter-router concept partially overlap with prior work (e.g., READ). Experiments are criticized for focusing on video/audio modalities instead of broader multi-modal tasks (e.g., autonomous driving), and the router’s theoretical justification is deemed underdeveloped. While authors address some gaps (e.g., sensitivity analysis, CMU-MOSI/MOSEI results), the distinction from READ remains unclear, and improvements in some cases are marginal.

The work’s focus on a critical real-world problem and empirical validation justifies its contribution.